# Remote land use impacts on river flows through atmospheric teleconnections

Lan Wang-Erlandsson[1,2,3], Ingo Fetzer[1], Patrick W. Keys[1,4], Ruud J. van der Ent[2,5], Hubert H. G. Savenije[2], and Line J. Gordon[1]

[1]Stockholm Resilience Centre, Stockholm University, Kräftriket 2B, 104 05, Stockholm, Sweden
[2]Department of Water Management, Faculty of Civil Engineering and Geosciences, Delft University of Technology, PO box 5048, 2600 GA Delft, The Netherlands
[3]Research Institute for Humanity and Nature (RIHN), 457-4 Motoyama, Kamigamo, Kita-ku, 603-8047 Kyoto, Japan
[4]School of Global Environmental Sustainability, Colorado State University, Fort Collins, CO 80523, USA
[5]Department of Physical Geography, Faculty of Geosciences, Utrecht University, P.O. Box 80 115, 3508 TC Utrecht, The Netherlands

*Correspondence to:* Lan Wang-Erlandsson (lan.wang@su.se)

**Abstract.** The effects of land-use change on river flows have usually been explained by changes within a river basin. However, land-atmosphere feedback such as moisture recycling can link local land-use change to modifications of remote precipitation, with further knock-on effects on distant river flows. Here, we look at river flow changes caused by both land-use change and water use within the basin, as well as modifications of imported and exported atmospheric moisture. We show that in some of the world's largest basins, precipitation was influenced stronger by land-use change occurring outside than inside the basin. Moreover, river flows in several non-transboundary basins was considerably regulated by land-use changes in foreign countries. We conclude that regional patterns of land-use change and moisture recycling are important to consider in explaining runoff change, integrating land and water management, and informing water governance.

## 1   Introduction

River flows ($Q$) are fundamental for ecosystems, nutrient transport, hydropower, navigation, and human well-being (Oki and Kanae, 2006). Land-use change (LUC) has been suggested to be the most important driver of both past (Piao et al., 2007; Sterling et al., 2012) and future (Betts et al., 2015; Milly et al., 2005) changes in river flows ($\Delta Q$). Central to the analysis of $Q$ is the river basin unit, and estimates of $\Delta Q$ from LUC often assume that impacts occur exclusively within a basin (Gerten et al., 2008; Piao et al., 2007; Rost et al., 2008a, b; Sterling et al., 2012). Water governance is strongly focused on frameworks such as the Integrated River Basin Management (IWRM) and largely assumes that there is no land-atmosphere feedback, even in discussions of spatial misfit between institutions and hydrological realities (Hoekstra, 2010; Giordano et al., 2015). In fact, land-atmosphere feedbacks are not incorporated in most recent literature on a wide range of topics of relevance for water management, such as virtual water (Dalin et al., 2017), the freshwater planetary boundary (Rockström et al., 2009; Steffen et al., 2015), water scarcity (Mekonnen and Hoekstra, 2016), relative role of climate and LUC for water flows (Zheng et al., 2016), and land acquisition impacts on water (Johansson et al., 2016; Rulli et al., 2012).

However, studies on land-atmosphere interactions clearly shows that changes in land surface properties can considerably influence precipitation ($P$) and $Q$ through land-atmosphere feedback, sometimes well beyond the local scale (Badger and Dirmeyer, 2016; Garcia et al., 2016; Avissar and Werth, 2005). For example, general circulation model simulations suggest that complete deforestation of Central Africa may decrease February $P$ by 35 % in the Great Lakes region (Avissar and Werth, 2005), and irrigation in India may support up to 40 % of the $P$ in some arid regions in Eastern Africa (de Vrese et al., 2016). Under a business-as-usual deforestation scenario, $Q$ in the Xingu river basin in the Amazon was found to increase by 10-12 % without land-atmosphere feedback, and decrease by 30-36 % when such feedback was taken into account (Stickler et al., 2013). Furthermore, statistical analyses of observed data suggests that irrigation in the US high plains enhances downwind $Q$ (Kustu et al., 2011) and coupled regional climate modeling shows that irrigation in the California Central Valley can be linked to about 30 % increase in Colorado $Q$ (Lo and Famiglietti, 2013). At the global scale, $\Delta Q$ from future climate and LUC scenarios changed from decrease to increase by considering land-atmosphere feedback and by closing the water balance (Betts et al., 2015).

Land-atmosphere interactions can influence $Q$ through thermal layer processes, terrestrial moisture recycling (TMR), and circulation perturbation (Goessling and Reick, 2011). First, thermal layer processes refer to the boundary layer and mesoscale circulation perturbation that may lead to a change in total terrestrial evaporation ($E$) and can locally lead to both positive and negative $P$ responses (Guillod et al., 2015; Seneviratne et al., 2010; Koster et al., 2003). Local forest clearing has for example been shown to enhance $P$ in downwind areas due to turbulence changes (Khanna et al., 2017; Saad et al., 2010). Second, TMR refers to the process of terrestrial $E$ returning to land as $P$ and is underpinned by the mass conservation of water (Brubaker et al., 1993). TMR is often the dominating land-atmosphere process at the regional to continental scale (D'Almeida et al., 2007; Spracklen et al., 2012; Lawrence and Vandecar, 2014; Tuinenburg, 2013). About 40 % of global terrestrial $P$ (van der Ent et al., 2014) originates from terrestrial $E$ and the average distance traveled in the atmosphere is 500-5000 km (van der Ent and Savenije, 2011) – a distance likely to exceed the size of most river basins. Lastly, large-scale atmospheric circulation perturbation allow extreme LUC (e.g., complete tropical deforestation) to impact $P$ in geographically remote regions and continents in unexpected ways (Avissar and Werth, 2005; Badger and Dirmeyer, 2016; Garcia et al., 2016; Lawrence and Vandecar, 2014). Monsoon regions are particularly sensitive to circulation perturbation, and irrigation may for example reduce $P$ by weakening the monsoon onset (Tuinenburg, 2013).

The previous studies that illustrated the importance of remote LUC for basin $P$ and $Q$, did not examine the effect of taking moisture recycling into account for estimating LUC effects on $Q$, nor analysed the interplay between LUC within and outside the river basin. These effects are, however, important to disentangle since they can have profound water governance implications for for example riparian water rights and transboundary river basin treaties (Keys et al., 2017; Dirmeyer et al., 2009; Ellison et al., 2017). Thus, there is a missing interdisciplinary bridge between understanding the role of land-atmosphere feedback over large distances and its importance for water governance at the basin scale.

This study aims to (*i*) investigate the potential impacts of human LUC on $Q$ worldwide accounting for TMR, (*ii*) disentangle the relative influence on $Q$ from within- and extra-basin LUC, (*iii*) attribute potential human LUC impacts on $Q$ to nation states, and (*iv*) discuss the potential implications for water governance. We focus on the TMR effect because it is transparent,

closes the water balance, and explicitly links changes in land and water geographically. Given these advantages, similar TMR approaches have in recent years been used to analyse unexplored relations, e.g., LUC impacts of crop yields (Bagley et al., 2012), self-amplifying forest die-back from TMR changes (Zemp et al., 2017), and vulnerability to LUC induced reductions in $P$ (Keys et al., 2016; Miralles et al., 2016). For a comparison of different methods for analysing LUC impacts on $Q$, see
Table S1.

## 2   Methods

### 2.1   Modelling

#### 2.1.1   Hydrological modelling

We used the process-based hydrological model Simple Terrestrial Evaporation to Atmosphere Model (STEAM) (Wang-Erlandsson
et al., 2014) to simulate water fluxes based on land cover and land use. STEAM partitions evaporation into five fluxes: vegetation interception, floor interception, transpiration, soil moisture evaporation, and open-water evaporation. STEAM uses the Penman-Monteith equation (Monteith, 1965) to estimate potential evaporation, the Jarvis-Stewart equation (Stewart, 1988) to compute stomatal resistance, and Jolly's growing season index (function of minimum temperature, soil moisture content, and daylight) to describe phenology (Jolly et al., 2005). STEAM operates at $1.5°x1.5°$ and a 3 hour resolution. Based on the
long term water balance, mean annual river flow ($Q$) is assumed to approximately equal the difference between mean annual precipitation ($P$) and evaporation ($E$), i.e., $Q = P - E$. STEAM was validated in previous studies (Wang-Erlandsson et al., 2014, 2016) and compared well with recent observation based analyses of evaporation partitioning by land-cover type (Wei et al., 2017). Modifications from the original version of STEAM (Wang-Erlandsson et al., 2014, 2016) includes: (1) update of land-use classification, parameterisation, and parametrisation approach (Table S2, and Fig. S1), (2) use of a temperature
threshold of 0 °C for differentiating snowfall from rainfall, and (3) differences in input data (i.e., root zone storage capacity, land surface map, precipitation data source as described in Data). Evaluation against runoff data is shown in Fig. S2. Simulated land-use change effects on evaporation increase and decrease are compared with literature values in Table S3 and found to be in the conservative range. With the study period being 2000–2013, the years 1995–1999 were used as spin-up for STEAM.

#### 2.1.2   Moisture tracking

Atmospheric moisture is tracked using the Eulerian moisture tracking scheme Water Accounting Model-2 layers (WAM-2layers) (van der Ent, 2014; van der Ent et al., 2014). WAM-2layers tracks atmospheric moisture from zero pressure to surface pressure in two layers. Within the layers, atmosphere is assumed to be well-mixed. WAM-2layers tracks vapor flows by ap-

plying the water balance. For example, the following equation is used to track where evaporation from a given region falls as precipitation (i.e., forward tracking):

$$\frac{\partial S_{\text{tracked}}}{\partial t} = \frac{\partial \left( S_{\text{tracked}} u \right)}{\partial x} + \frac{\partial \left( S_{\text{tracked}} v \right)}{\partial y} + E_{\text{tracked}} - P_{\text{tracked}} \pm F_{\text{vertical,tracked}} \tag{1}$$

where $S_{\text{tracked}}$ is the tracked atmospheric storage in an atmospheric column in one layer, $t$ is time, $u$ and $v$ are wind compo-
nents in the $x$ zonal and $y$ meridional direction, $E_{\text{tracked}}$ is tracked evaporation entering and $P_{\text{tracked}}$ is precipitation exiting an atmospheric column and layer, and $F_{\text{vertical,tracked}}$ is the tracked vertical moisture transport between the two layers. An analogous equation is used for tracking the source of precipitation to a given region (i.e., backward tracking). The spatial resolution of WAM-2layers is $1.5°$ and input data are linearly interpolated to the 15 minute time step to maintain numerical stability. WAM-2layers has been employed previously for analysing atmospheric moisture transport over terrestrial areas (Keys et al.,
2012, 2016) and validated against other types of moisture tracking algorithms (van der Ent et al., 2013). We used the MATLAB (The MathWorks, 2014) version of WAM-2layers, but a Python version is also openly available on Github (van der Ent, 2016). With the study period being 2000–2013, the year 1999 is used as spin-up in forward tracking in WAM-2layers, and 2014 is used as spin-up for backward tracking in WAM-2layers.

### 2.1.3 Coupling of the moisture tracking scheme and the hydrological model

Hydrological flows in the current land-use scenarios is simply represented by current data and simulation. To obtain $E$ and $P$ under potential land cover, STEAM is coupled with WAM-2layers by (1) simulating present day $E$ in STEAM and forward tracking terrestrial $E$ with WAM-2layers, meaning that the $E_{\text{tracked}}$ is equal to all evaporation from terrestrial surfaces, i.e., not belonging to the oceans, (2) simulating $E$ in STEAM based on present day $P$ and potential land cover, and forward track the fate of terrestrial $E$ with WAM-2layers, (3) calculating the change in $P_{\text{tracked}}$, (4) updating the present day $P$ with the changes in $P_{\text{tracked}}$, and (5) simulating $E$ in STEAM based on updated $P$ and potential land cover, and forward tracking the fate of terrestrial $E$ with WAM-2layers, see Fig. 1. Steps 3–5 are iterated until the annual $P$ change is below $1\%$ and the monthly $P$ change is below $5\,\text{mm month}^{-1}$ in every grid cell, which in our case ultimately resulted in four iterations in total. This procedure assumes that land-use induced changes in terrestrial $E$ will result in proportional changes in $P$ with terrestrial origin.

### 2.2 Data

#### 2.2.1 Land data

Land use and land cover data input to STEAM are based on the Ramankutty potential land-cover (Ramankutty and Foley, 1999) and current land-use scenarios (Ramankutty et al., 2008) for consistency. We further added permanent wetlands, permanent snow or ice, and urban or built-up areas from the Land Cover Type Climate Modeling Grid (CMG) MCD12C1 International Geosphere Biosphere Program (IGBP) land classification created from Terra and Aqua Moderate Resolution Imaging Spec-

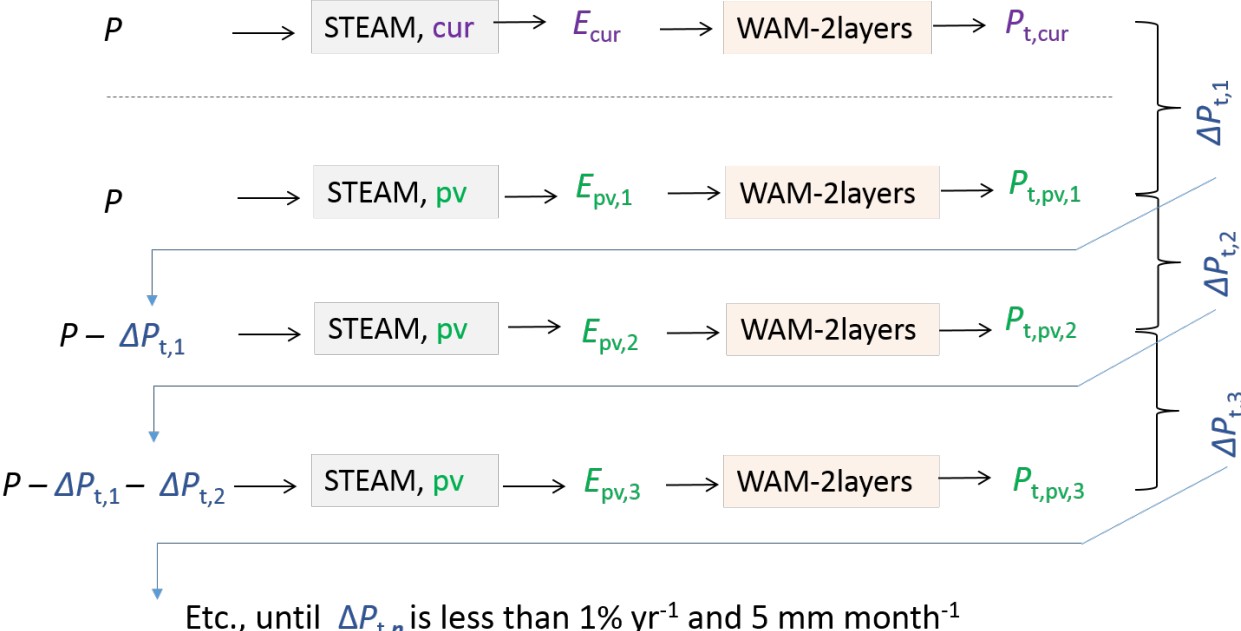

**Figure 1.** Model coupling schematic. Model coupling between STEAM and WAM-2layers based on current land use and potential vegetation scenarios. $P$ stands for current precipitation, $E$ stands for evaporation. Subscript t stands for terrestrial origin, pv denotes simulation with potential vegetation, cur denotes simulation with current land use, and $n$ stands for the number of iterations.

troradiometer (MODIS) data (Friedl et al., 2010) for the year 2005. Monthly irrigated rice and irrigation non-rice crops were obtained from the data set of Monthly Irrigated and Rainfed Crop Areas around the year 2000 (MIRCA2000) V1.1. (Portmann et al., 2010). The urban and irrigated areas were only added to the current land cover map. In this merging procedure, MODIS is allowed to override the Ramankutty datasets, and MIRCA2000 is allowed to override the Ramankutty map as long as it does
5    not extend over the cropland areas. The scenarios used are shown in Fig. S3 and the land-use change is illustrated in Fig. 2.

The root zone storage capacity map is based on a climate-observation based root zone storage capacity ($S_R$) (Wang-Erlandsson et al., 2016) derived from satellite and energy balance based evaporation, gauge-based precipitation, and modelled irrigation. The best performing Gumbel normalised root zone storage capacity ($S_{R,CRU-SM,merged}$) was used. Root zone storage capacity for both current and potential land-cover and land-use scenarios were constructed from mean of land-cover type and
10   Köppen-Geiger climate class (Kottek et al., 2006). The mean root zone storage capacity of single land-cover types was used only in places where the combination of land-cover type and climate zone that exists in the potential land-cover scenario did not exist in the current land-use map.

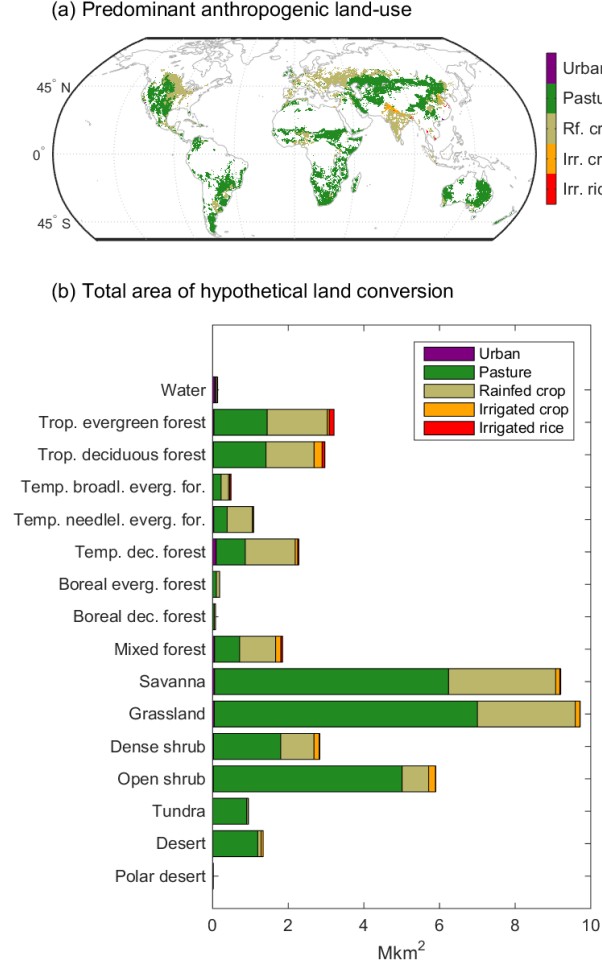

(a) Predominant anthropogenic land-use

(b) Total area of hypothetical land conversion

**Figure 2.** Changes in land use resulting from the replacement of the potential vegetation scenario with the current land-use map. Changes in **a**, land use (current land use is shown, with grid cells without major land-use change masked out), **b**, total area of difference between potential vegetation (y-axis) and current land-use map (colour legend).

### 2.2.2   Meteorological forcing and runoff data

Meteorological data used in WAM-2layers and STEAM, except for land precipitation, were taken from the Earth Retrospective Analysis Interim (ERA-I) from the European Centre for Medium-Range Weather Forecasts (ECMWF) (Dee et al., 2011). ERA-I meteorological forcing to STEAM are: snowmelt, temperature at 2 m height, dew point temperature at 2m height, wind speed (meridional and zonal vectors) at 10 m height, incoming shortwave radiation, and net longwave radiation. In addition, ERA-I evaporation data were used to downscale calculated daily potential evaporation in STEAM to the 3 hour time step. ERA-I model level forcings used in the WAM-2layers are specific humidity, and wind speed at 6 hourly resolution, spanning from zero to surface pressure. Moreover, 3 hourly ocean evaporation is taken from ERA-I. The Modern-Era Retrospective analysis for Research and Applications (MERRA) reanalysis has in a previous study been used as input to WAM-2layers for comparison and generated similar persistent moisture recycling patterns, except in South America where differences arise due to underestimation of precipitation in MERRA (Keys et al., 2014). Precipitation forcing for WAM-2layers and STEAM both come from the state-of-the-art product Multi-Source Weighted-Ensemble Precipitation (MSWEP V1) (Beck et al., 2017) that was specifically created for hydrological modeling. The use of MSWEP as forcing for STEAM resulted in runoff estimates that compare well to observed runoff data (Fig. S2). All meteorological forcing data cover temporally 1995—2014.

Runoff data used for benchmarking were taken from the composite (observed river discharge consistent with water balance model) from the Global Runoff Data Centre (GRDC) (Fekete et al., 2002). The separate GRDC water balance model runoff fields are included in the comparison for reference (Fig. S2). Where available in the literature, we also compare our simulated river flows to more reliable and recent discharge data in individual basins.

The spatial coverage of all data used is 57°S-79.5°N latitudes at 1.5° x 1.5° resolution. MSWEP originally at 0.25° and GRDC runoff at 0.5° were aggregated to 1.5° resolution by simple averaging.

### 2.3   Analyses

#### 2.3.1   Changes in hydrological flows

River flow change without TMR ($\Delta Q_{\mathrm{noTMR}}$) is

$$\Delta Q_{\mathrm{noTMR}} = (P_{\mathrm{cur}} - E_{\mathrm{cur}}) - (P_{\mathrm{cur}} - E_{\mathrm{pv,1}}) \tag{2}$$

where $P_{\mathrm{cur}}$ is current day precipitation data from MSWEP, $E_{\mathrm{cur}}$ is current day evaporation based on STEAM simulation, and $E_{\mathrm{pv,1}}$ results from STEAM simulation in the potential vegetation scenario and forced with current day precipitation (Fig. 1). River flow change after accounting for TMR ($\Delta Q$) is

$$\Delta Q = (P_{\mathrm{cur}} - E_{\mathrm{cur}}) - (P_{\mathrm{pv,4}} - E_{\mathrm{pv,5}}) \tag{3}$$

where $P_{\mathrm{pv,4}}$ is the converged precipitation (i.e., meeting the convergence requirement of mean annual precipitation change < 1 % and monthly precipitation change < 5 mm/month in every grid cell) achieved at the fourth iterative coupling between

STEAM and WAM-2layers, and $E_{pv,5}$ is the evaporation under the potential vegetation scenario simulated in STEAM with precipitation forcing $P_{pv,4}$.

Change in tracked basin precipitation ($\Delta P_{\text{tracked,basin}}$) occurring *outside* the river basin boundaries is referred to as $\Delta P_{\text{import}}$, whereas $\Delta P_{\text{tracked,basin}}$ originating from *within* the basin boundary is referred to as $\Delta P_{\text{basin-recycling}}$. Internally recycled evaporation ($\Delta E_{\text{basin-recycling}}$) corresponds to $\Delta P_{\text{basin-recycling}}$ and all other basin evaporation change is considered exported ($\Delta E_{\text{basin,recycling}}$).

### 2.3.2   Country influence on changes in river flows

The influence on river flow change in river basin b from country c without considering TMR ($I_{b,c,\text{noTMR}}$) is:

$$I_{b,c,\text{noTMR}} = |\Delta E_{b,c}|. \tag{4}$$

where $\Delta E_{b,c}$ is evaporation change in the part of river basin b located in country c. The influence on river flow change in basin b from country c with consideration of TMR ($I_{b,c,\text{TMR}}$) is:

$$I_{b,c,\text{TMR}} = |\Delta E_{b,c,\text{export}}| + |\Delta P_{b,c,\text{import}}|) \tag{5}$$

where $\Delta E_{b,c,\text{export}}$ is the evaporation change exported from the part of basin b located in country c, and $\Delta P_{b,c,\text{import}}$ is the precipitation change imported to basin b from country c.

Influences from countries below 5 % of total influences in a specific basin ($I_{b,c,\text{noTMR}} < 0.05 \times \sum I_{b,c,\text{TMR}}$) were lumped into the category "Other".

## 3   Results

### 3.1   LUC impacts on global water flows

Our results show that human LUC (from potential land cover to current land use) (Fig. 2) has led to reductions in $E$ and $P$, and to increases in $Q$, in large parts of the world (Fig. 4b–d). $E$ has decreased primarily in Southwest China, Europe, West Africa, south of Congo, and southeast South America resulting from substantial pasture and agricultural expansion (Ramankutty et al., 2008). Following prevailing wind directions (Fig. 3c), subsequent $P$ has decreased in all tropical regions, in South Central China, eastern US, and Europe.

Nevertheless, in some areas, $E$ increased due to incremental irrigation — notably in India, the Western US, Northeast China, and in the Middle East (Fig. 4a,b). Due to the combination of heavy irrigation in India and orography, $P$ has increased substantially along the Himalaya mountain ridge (Fig. 4b,c). Weak increases in $P$ are observed in other downwind regions: the Sahel (i.e., downwind irrigation areas along the Nile) and in the Western US. Continental precipitation recycling ratios are modified — with some exceptions — in a similar pattern as $P$ (Fig. 4e,f). Large $\Delta Q$ are seen in the La Plata basin in South America, the Zambezi in Southern Africa, the Yangtze in China, and the Indus in North India (Fig. 4g), and relative changes in $Q$ are large in for example the Colorado basin in the US, the Odra basin in eastern Europe, and Lake Chad river basin in Africa (Fig. 4g).

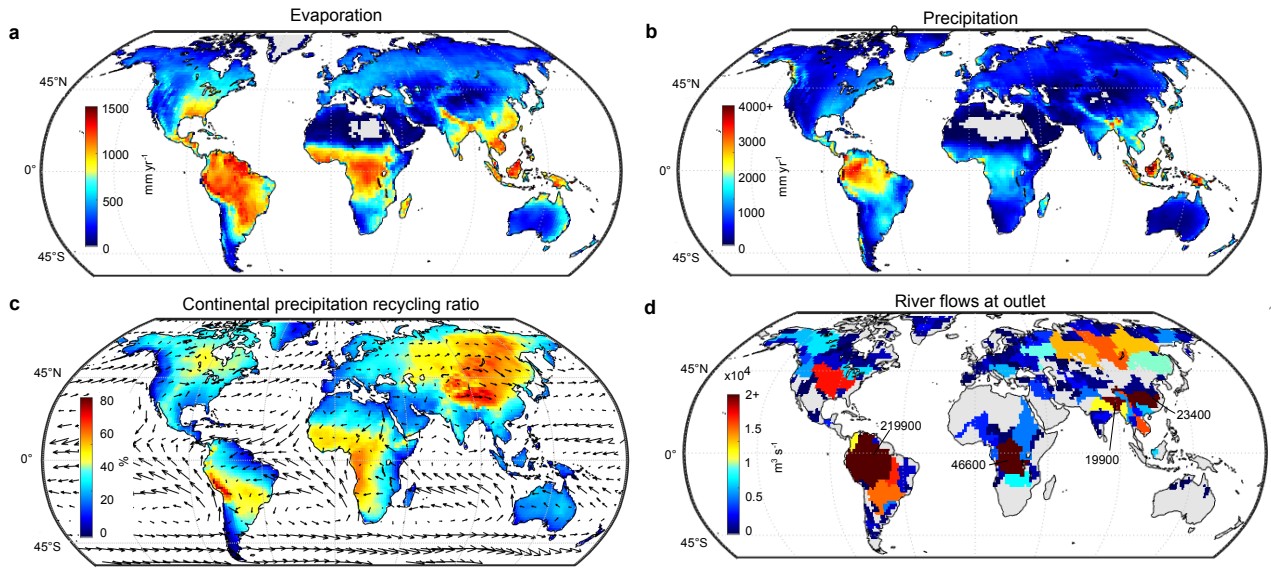

**Figure 3.** Current mean annual hydrological flows 2000–2013. **a**, Current evaporation simulated by STEAM, **b**, current precipitation (MSWEP data), **c**, current continental precipitation recycling ratio (i.e., precipitation with terrestrial origin divided by total precipitation: $P_{\text{tracked}}/P$) where arrows show average winds in the lower atmosphere, **d**, current river flow at outlet based on $P$-$E$. Values below about 0.5 % of maximum display value are in grey.

## 3.2 The role of TMR for $\Delta Q$

In aggregate (Fig. 5), when accounting for TMR, LUC changed global terrestrial $E$ by $-1251$ km$^3$ yr$^{-1}$ ($-1.8$ % from 69,211 km$^3$ yr$^{-1}$), $P$ by $-586$ km$^3$ yr$^{-1}$ ($-0.5$ % from 107,800 km$^3$ yr$^{-1}$), and $Q$ by 664 km$^3$ yr$^{-1}$ ($+1.7$ % from 38,589 km$^3$ yr$^{-1}$). The estimated changes to $Q$ tend to fall in the conservative end of previous estimates (Gerten et al., 2008; Piao et al.,

2007; Rost et al., 2008a, b; Sterling et al., 2012) (Fig. 5). However, recent research (Jaramillo and Destouni, 2015) suggests that consumptive water use is severely underestimated in earlier studies (e.g., Döll et al., 2009; Sterling et al., 2012). $\Delta Q$ with TMR corresponds to the difference between $\Delta E$ and $\Delta P$ change including TMR (Fig. 5, solid bars), whereas $\Delta Q$ without accounting for TMR simply corresponds to $\Delta E$ without TMR (Fig. 5, hollow bars).

Including TMR nearly halves the global $\Delta Q$ estimate. This is because $E$ returns as $P$ over land and thus compensates for the

initial water "loss" from the basin. This suggests that previous studies without TMR (e.g., Gerten et al., 2008; Piao et al., 2007; Sterling et al., 2012) may have substantially overestimated the net LUC impacts on $Q$. Our estimate of LUC impact on $Q$ is slightly larger than some of the estimates of $CO_2$ fertilisation (e.g., Alkama et al., 2010; Gerten et al., 2008), but substantially smaller than climate change and overall human impact (e.g., Alkama et al., 2010; Gerten et al., 2008) (Fig. 5).

Our river basin analysis shows that accounting for TMR considerably alters estimates of $\Delta Q$ (Fig. 7a): in the Congo, Volga,

and Ob basins, $\Delta Q$ are reduced by more than half; in the Amazon, $\Delta Q$ drops from 1630 to 270 m$^3$ s$^{-1}$; and in the Yenisei, the sign of $\Delta Q$ is reversed from an increase (150 m$^3$ s$^{-1}$) to a decrease ($-220$ m$^3$ s$^{-1}$).

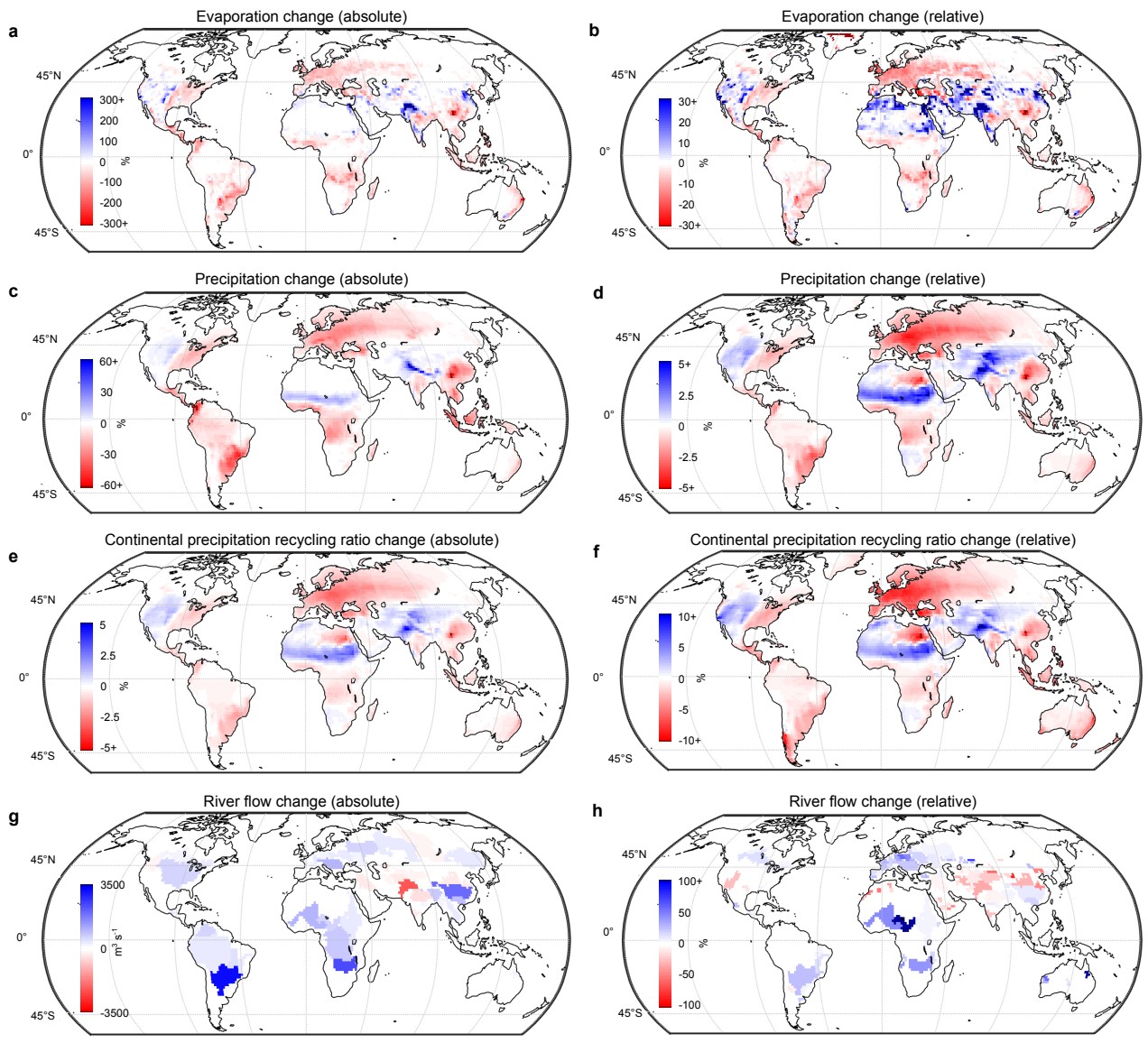

**Figure 4.** Land-use change induced changes in hydrological flows (current land-use - potential vegetation scenario): **a**, absolute change in evaporation, **b**, relative change in evaporation, **c**, absolute changes in precipitation, **d**, relative change in precipitation, **e**, absolute change in continental precipitation recycling ratio (i.e., precipitation with terrestrial origin divided by total precipitation $P_{tracked}/P$ and converted to the unit of percent), **f**, relative change in continental precipitation recycling ratio, **g**, absolute change in river flow at outlet, and **h**, relative change in river flows at outlet.

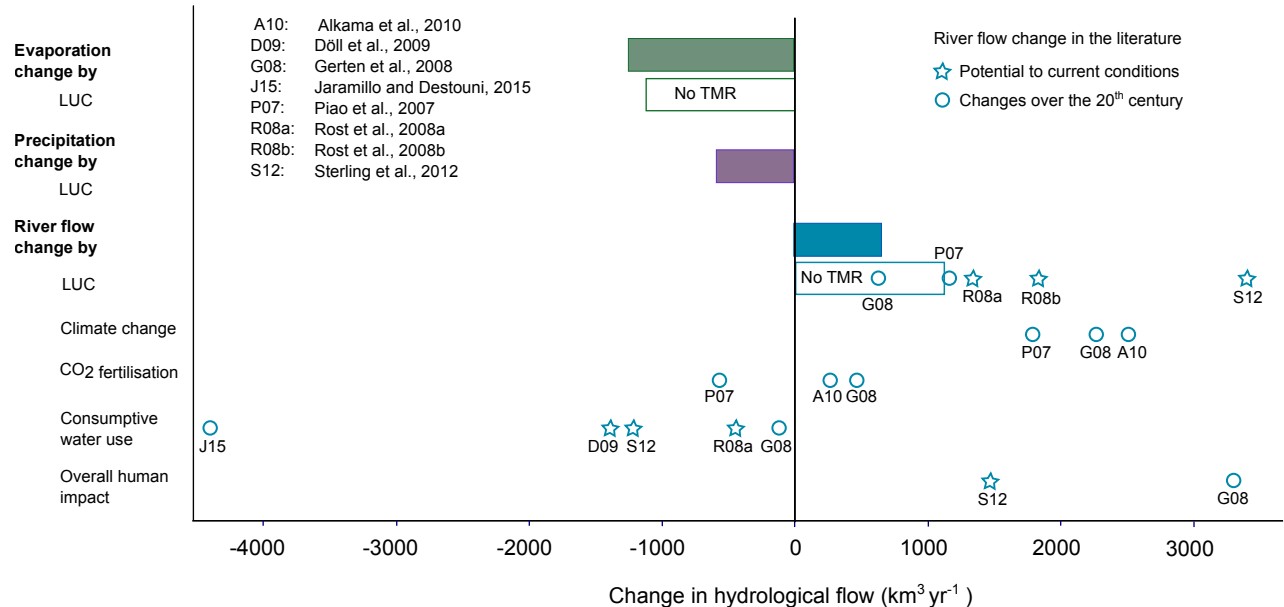

**Figure 5.** Human impact on global hydrological flows. The solid bars show our estimated net change (terrestrial area 131.7 106 km$^2$ and time period 2000-2013) in evaporation, precipitation and river flows including consideration of terrestrial moisture recycling (TMR). Hollow bars show flow changes without TMR. Circles and stars indicate river flow change estimates from other studies (Table S4), where land-use change (LUC) implicitly accounts for consumptive water use. Note that while consumptive water use alone always reduces river flows, other human impacts have both positive and negative influences that are concealed by the global aggregate.

At the basin level, the TMR effect on river flow change is estimated to be the largest in large and relatively wet basins such as the Amazon, Congo, and Yangtze river basins in terms of absolute volumes (Fig. 6a). Not accounting for TMR clearly generates the largest relative deviations in river flow change estimates in the Amazon (i.e., $\Delta Q_{\mathrm{noTMR}}$ is approximately five times larger than $\Delta Q$), and large relative TMR effects are seen in many large basins worldwide, including e.g., Congo ($\Delta Q_{\mathrm{noTMR}}$ is 150 %

5 higher than $\Delta Q$), Yenisei ($\Delta Q_{\mathrm{noTMR}}$ is 165 % lower than $\Delta Q$), and Ob ($\Delta Q_{\mathrm{noTMR}}$ is 140 % higher than $\Delta Q$) river basins (Fig. 6b). The TMR effect relative $Q_{\mathrm{cur}}$ (Fig. 6c) shows that TMR effect can be important also in more arid basins such as the Colorado, Niger, and the Yellow river.

## 3.3 The interplay between internal and external LUC

Furthermore, atmospheric moisture does not respect river basin boundaries (Fig. 7a., and spatial maps in Fig. S4, S5, S6, and

10 S7). In fact, $P$ over the basins has been modified more significantly by external than by internal LUC (change in imported precipitation $\Delta P_{\mathrm{import}}$ > change in internally recycled precipitation $\Delta P_{\mathrm{basin-recycling}}$) in some of the largest basins (Fig. 7a). Likewise, internally recycled evaporation changes ($E_{\mathrm{basin-recycling}}$) (Fig. 7b, II) are substantially smaller than $\Delta E$ affecting $P$ elsewhere ($\Delta E_{\mathrm{basin-recycling}}$ < change in exported evaporation $\Delta E_{\mathrm{export}}$) for all selected river basins (Fig. 7a).

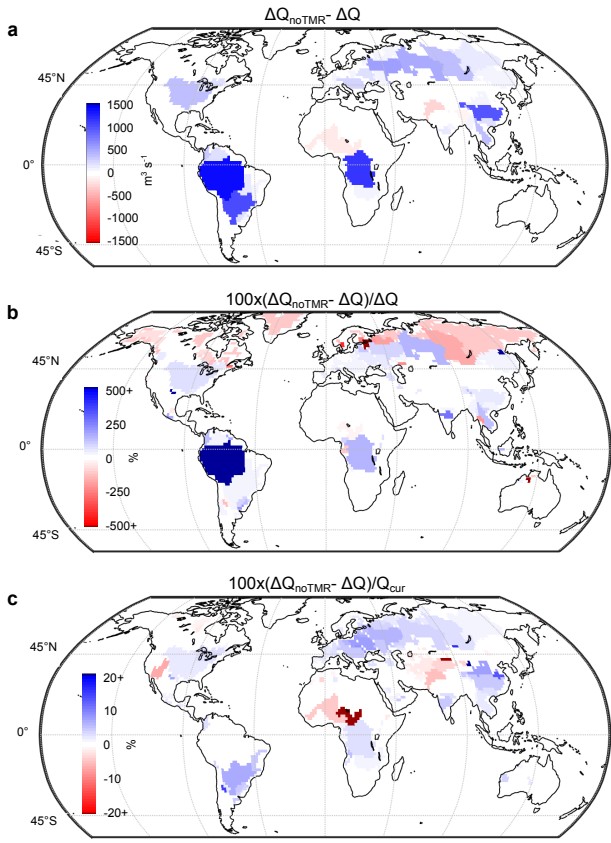

**Figure 6.** The effect of accounting for TMR on river flow change estimates, shown **a**, as absolute difference between river flow change without and with TMR effect, i.e., $\Delta Q_{\text{noTMR}}$ - $\Delta Q$, **b**, as this difference relative river flow change with TMR effect, i.e., $(\Delta Q_{\text{noTMR}}$ - $\Delta Q)/\Delta Q$), and **c**, this difference relative current river flows, i.e., $(\Delta Q_{\text{noTMR}}$ - $\Delta Q)/Q_{\text{cur}}$

Internal moisture recycling (Fig. 7b, II) does not affect $\Delta Q$ directly, only indirectly if $\Delta P_{\text{basin}-\text{recycling}}$ affects subsequent $\Delta E_{\text{export}}$ under transient change (Fig. 7b and Methods). Thus, provided steady-state, $\Delta Q$ simply corresponds to the difference between $\Delta E_{\text{export}}$ and $\Delta P_{\text{import}}$ (Fig. 7a). For example, $\Delta Q$ in the Amazon is very small because the reduced $\Delta P_{\text{import}}$ is almost entirely offset by reduced $\Delta E_{\text{export}}$. In Congo, about half of the within-basin LUC induced $Q$ increase is counteracted
5 by extra-basin LUC (i.e., $\Delta P_{\text{import}} \approx 0.5 \Delta E_{\text{export}}$). The effect of TMR on $\Delta Q$ ($\Delta Q_{\text{noTMR}} - \Delta Q$, where subscript $_{\text{noTMR}}$ denotes simulation without TMR) corresponds to total $\Delta P$ (i.e., $\Delta P_{\text{import}} + \Delta P_{\text{basin}-\text{recycling}}$) and any indirect $\Delta E$ (i.e., $\Delta E_{\text{noTMR}} - \Delta E$, not shown). In Yangtze, the $\Delta Q$ is mitigated mostly by $\Delta P_{\text{basin}-\text{recycling}}$. The strong flow reduction in the heavily irrigated Indus, however, is only mildly compensated by TMR (i.e., $\Delta P_{\text{import}} \ll \Delta E_{\text{export}}$).

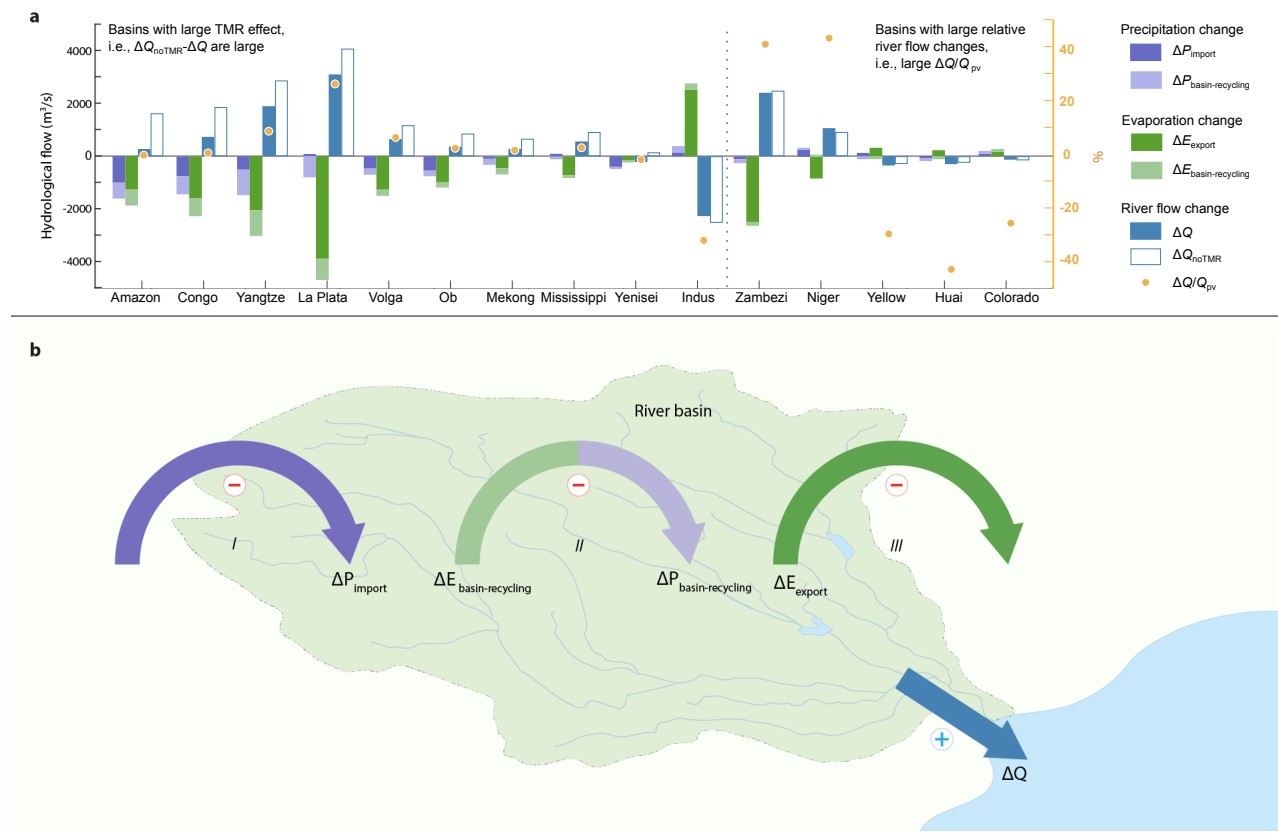

**Figure 7.** Changes in hydrological flows at river basin scale. **a**, Changes in hydrological flows by ten of the basins with the largest terrestrial moisture recycling (TMR) effect on river flows ($Q$) (eight basins with increased and two with decreased $Q$), and five basins with large relative changes in river flows (two basins with positive $\Delta Q$ and three with negative $\Delta Q$). **b**, Conceptual figure of hydrological flow changes in a basin. The (-) and (+) in b may be different for different basins, and the (-) and (+) as displayed here are for example seen in Amazon, Congo, and Yangtze, see a. Note also that the figure has two y-axis, $\text{m}^3\,\text{s}^{-1}$ to the left and % to the right

## 3.4 Attributing influence on $\Delta Q$ to nations

Typically, TMR attributes LUC influence on $Q$ to a larger number of nations than when only basin boundaries are considered (Fig. 8 and Fig. S8). For example, in the Amazon, $\Delta Q$ originates from as far away as Africa if considering TMR (Fig. 8b-c). Because of the large LUC-induced $E$ reductions, the African influence on Amazon $\Delta Q$ is comparable to within-basin influence. Notably, basins geographically confined within one nation can be influenced by LUC taking place in foreign nations. This is for example the case in Yangtze, where simulated irrigation in India increases the basin's $P$ (Fig. 8g).

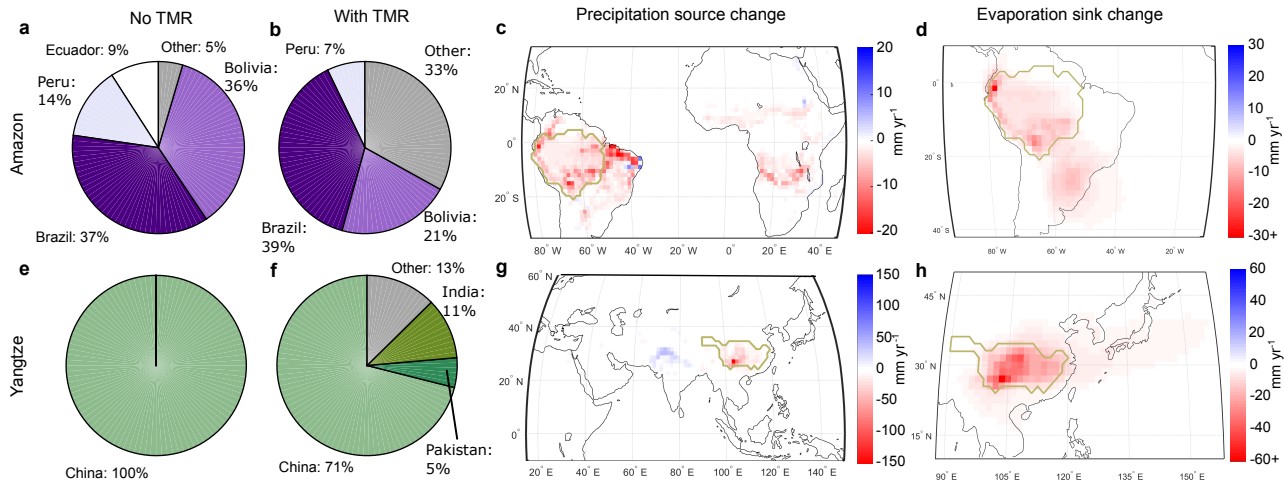

**Figure 8.** Nation influence of river flow change depending on consideration of terrestrial moisture recycling (TMR) in **a–b**, Amazon and **e–f**, Yangtze. Without TMR, the nation influence on river flow change originates entirely from within-basin country-wise evaporation change (**a**, **e**). By considering TMR, the nation influence **b**, **f** is the sum of absolute imported precipitation change (precipitation source outside encircled basin boundaries in **c** and **g**) and of absolute exported evaporation change (evaporation sink outside encircled basin boundaries in **d** and **h**) (Methods). Precipitation sources (**c**, **g**) and evaporation sinks (**d**, **h**) within basin boundaries are recycled.

## 4 Discussion

### 4.1 Interplay between TMR and LUC

At the global scale, $\Delta Q$ as a response to LUC can be almost halved by taking TMR into account (Fig. 5). However, these effects vary widely by regions. While the TMR effects are negligible in some basins, remote LUC can compensate the majority

5 of the impact on $Q$ from local LUC in other basins (e.g., Amazon, Fig. 7a) and even propose new transboundary relationships (e.g., Yangtze, Fig. 8e). From a TMR perspective, the impact on $Q$ from within-basin LUC depends on the $\Delta E$ exported from the basin as much as the $\Delta P$ imported to the basin.

Our analysis shows the importance of considering LUC on par with TMR to identify anthropogenic influence on water resources, beyond analyses of pure moisture exchanges (Dirmeyer et al., 2009; Keys et al., 2017). While Africa does not

10 constitute a major moisture source of Amazonian $P$ (7 % of all Amazon $P$, 13 % of Amazon $P$ with continental origin, see also Fig. S4a), the spatial extent of $\Delta E$ from LUC was sufficient to elevate the relative importance of African LUC for Amazonian $\Delta Q$ (28 % of Amazon $\Delta P$, see also Fig. 8c). Similarly, India is not identified as a major moisture source of Yangtze (see Fig. S4c and Wei et al. (2012), but has about 10 % influence on Yangtze $\Delta Q$ (Fig. 8f and 8g).

## 4.2 Potential governance relevance

Our results indicate that both precipitation- and evaporationsheds of river basins are relevant governance units. Previous studies of TMR for water management (Berger et al., 2014; Keys et al., 2017) have emphasized the importance of considering the $P$ source region, i.e., the precipitationshed (Keys et al., 2012), which was introduced as a concept analogue to watershed for water resources management. This study finds that the evaporationshed (van der Ent and Savenije, 2013), i.e., the $E$ sink region, is just as important when considering changes to $Q$.

LUC impacts $Q$ through TMR in different ways depending on how precipitationshed, river basin, and evaporationshed are aligned. For example, where an evaporationshed has a limited overlap with river basin boundaries, reforesting a river basin may lead to unexpectedly large reductions in $Q$, if considerable deforestation simultaneously occurs in the precipitationshed outside the river basin.

The magnitude of TMR effects from remote LUC on $Q$ can be comparable to managed water flows. For example, Yangtze River provides 36 % of the country's surface water resources, and is subject to two of the world's most ambitious water engineering projects: the Three Gorges Dam and the South-to-North Water Diversion (CWRC, 2017). The overall TMR effect on mean annual LUC-induced $\Delta Q$ is here estimated at 980 m$^3$ s$^{-1}$ in the Yangtze basin, and the mean annual moisture change imported to the basin from foreign countries is estimated at 1,110 m$^3$ s$^{-1}$ (Fig. 8g). As comparison, the $Q$ difference between a normal and a dry year is about 300 m$^3$ s$^{-1}$ and the total amount of water to be transferred from the Yangtze through the South-to-North Water Diversion is aimed to be 1420 m$^3$ s$^{-1}$ (NSBD, 2011). Seasonal and interannual flow variability is a major challenge facing the Yangtze, and future research in the seasonal LUC influence and interaction with the monsoon system is needed. Note, however, that our estimates are associated with parameter sensitivity (see Fig. S9) and large uncertainties as discussed in the Limitations.

We note that the relevance of considering TMR governance depends on future LUC. The simulated $\Delta Q$ in this paper follows from a rather extreme LUC scenario (from potential to current land-use). The current LUC in this study is 15 million km$^2$ cropland and 28 million km$^2$ pasture conversion (Ramankutty et al., 2008). As comparison, models used in the Intergovernmental Panel on Climate Change (IPCC) Fourth Assessment Report (AR4) estimated cropland changes from $-1.2$ to $+12$ million km$^2$ between 2000 and 2050 (IPCC, 2007). A more recent multi-model comparison range cropland conversion until 2050 from $-1$ to $+8.5$ million km$^2$ across different scenarios (Schmitz et al., 2014). In total, the potential land for agricultural conversion has been estimated at 17 million km$^2$ (Schmitz et al., 2014). Thus, future LUC can be considerable, and potential TMR impacts on $Q$ will be dependent on the type and geographical distribution of LUC, as well as dependent on prevailing winds, hydroclimate, and orography.

## 4.3 Limitations

In interpreting our results, it should be noted that our approach only accounts for the TMR effects. The frequency or intensity of $P$ are assumed to remain unaffected by thermal layer processes or circulation perturbation, which may introduce a bias in the quantitative estimates of hydrological flows under water limited conditions (i.e., semi-arid regions and temperate region

during summertime) (Medvigy et al., 2011). Furthermore, vegetation response to $\Delta P$ is not simulated, such as forest dieback from increased fire risk under drying conditions. Human modification of $Q$ through dams and climate change (Haddeland et al., 2014) are also not considered in this study. In addition, the land-use change over land may affect above ocean processes mainly through modification of the energy balance and circulation in monsoon regions, which we do not account for. Changes in fresh water discharge to the oceans might have implications for ocean circulation and climate, as studies of for example river discharge to the Arctic Ocean showed (Peterson, 2002, 2006). However, moisture recycling's buffering effect (which mitigates river flow changes), should have a mitigating effect on the ocean's response to fresh water inflow. Otherwise, precipitation over ocean can influence ocean salinity (IPCC, 2013) and precipitation patterns over land can be influenced by sea surface temperature (Xie et al., 2010), but we consider this outside the scope of our study and likely to be of minor importance for the research questions that we address. Our TMR analyses should, thus, be seen as an inquiry to better understand the relative importance of local and remote LUC effects on $Q$ from a water balance perspective, rather than an exact prediction. Nevertheless, due to the inevitable recycling of moisture in the global hydrological cycle, uncertainties in the magnitude will unlikely affect our key conclusions that upwind extra-basin LUC can be essential for $Q$.

The magnitude of our estimated $\Delta P$ (Fig. 5) and $\Delta Q$ from LUC is conservative in comparison to the literature (Spracklen and Garcia-Carreras, 2015). For example, a meta-analysis of 96 different general circulation models (GCM) and regional climate model (RCM) deforestation simulations showed that under 10 % conversion of Amazon forest to pasture or soybean production, the inter-quartile range of rainfall change in the Amazon basin is 0 to $-4$ % (Spracklen and Garcia-Carreras, 2015). In comparison, the STEAM-WAM2layers approach with change from potential to current land-use change (i.e., 8.8 % deforestation extent in the Amazon), causes a rainfall reduction of 0.4 % in the Amazon and thus falls in the conservative range. In addition, our analyses concern mean annual $\Delta Q$, and can also be considered conservative in the sense that seasonal signals are expected to be much stronger.

The limitations of our methods should also been seen in light of the strengths and limitations of alternative methods for studying hydrological LUC effects, see Table S1. The most complex and coupled modelling approaches account for the highest number of feedback processes. However, the high degree of freedom in GCMs also contributes to the high sensitivity of precipitation to initial conditions and the low signal-to-noise ratios. For example, a scenario replacing natural with present-day land cover only detected a significant response in less than 5 % of all grid cells in a single model analysis (Findell et al., 2007) and less than 5 % in non-perturbed grid cells across seven different models (Pitman et al., 2009). Regional deforestation scenarios generate higher ratios of significant results near the source of change, albeit noise remains high in distant regions (Werth and Avissar, 2002). The challenges in simulating precipitation due to cloud formation, aerosol representation, and inherent uncertainties in circulation response (Aloysius et al., 2016; Koren et al., 2012; Shepherd, 2014), and non-closure of water balance in semi-coupled modelling approaches (Bring et al., 2015) also contribute to a high model dependence in estimates of river flow change from LUC (Kundzewicz et al., 2007). Thus, the sign, magnitude, and location of impacts vary widely among models (Aloysius et al., 2016; Pitman et al., 2009). Observation-based methods relate presence of vegetation or irrigation to precipitation or river flows using statistical methods, often in combination with moisture tracking to determine the geographical origin of rainfall (DeAngelis et al., 2010; Kustu et al., 2010, 2011; Spracklen et al., 2012). Limitations of this type

of methods include variations in data quality, challenges in isolating effects of land-use from climate variability, and difficulties establishing causation from correlation (Matin and Bourque, 2015). Key elements missing in all approaches including our own are socio-economic dynamics and landscape resilience, which are complex issues currently explored in experimental model settings (Nitzbon et al., 2017; Reyer et al., 2015).

## 4.4 Future research outlook

A key challenge for considering TMR effects in water governance is the modeling uncertainties and inherent variabilities associated with land-atmosphere feedback processes. The most complex modeling approaches account for the highest number of feedback processes. However, the sign, magnitude, and location of impacts vary widely even among state-of-the-art climate models (Pitman et al., 2009; Aloysius et al., 2016). Key future improvements in climate models' ability to simulate $\Delta P$ from LUC will contribute to the governability of TMR. In-depth examination of differences in model simulation of $P$ (e.g., the ongoing Precipitation Driver Response Model Intercomparison Project (Myhre et al., 2017)) is one step in this direction. Tracking moisture in coupled climate models could further help identify causes for simulated differences in atmospheric and hydrological outputs. Key elements missing in current research on LUC effects on hydrological flows include socio-economic dynamics and landscape resilience, which are complex issues currently explored in experimental model settings (Nitzbon et al., 2017; Reyer et al., 2015). In the meantime, "no-regret" policies in river basin management, where TMR objectives align with other aims can potentially be explored in conjunction with LUC scenarios that include TMR effects.

## 5 Conclusions

We analysed the potential impact of human LUC on $Q$ worldwide through TMR, and separately looked at the remote and local LUC effects of relevance to water governance. Despite the river basin being the standard unit in water governance and water resources management, we find that $\Delta Q$ are ultimately dependent on the modifications in both incoming $P$ and outflowing $E$. For example, where extra-basin LUC affects basin $P$ more strongly than within-basin LUC, reforesting a river basin may lead to unexpectedly large reductions in $Q$ if deforestation simultaneously occurs in $P$ source regions outside the river basin. Therefore, we emphasize the necessity of considering both the origin of basin $P$ as well as the fate of basin $E$ for management of local water resources. Further, we suggest the potential need for transboundary governance of river basins where extra-basin LUC is important for $\Delta Q$. International governance arrangements of teleconnnected LUC influence could be needed, even for river basins that today are not considered transboundary. We conclude that consideration of TMR is essential for understanding $Q$ modifications and managing water resources in a rapidly changing and tele-coupled world (Liu et al., 2013) facing increasing pressure on both land (Schmitz et al., 2014) and water (Mekonnen and Hoekstra, 2016). Further research in both climate modeling and water governance strategies is needed to internalize land-atmosphere interactions in future water resources considerations.

*Code and data availability.* The moisture tracking scheme Water Accounting Model-2 layers (WAM-2layers) in Python code can be obtained from GitHub (https://github.com/ruudvdent/WAM2layersPython). Earth Retrospective Analysis Interim (ERA-I) meteorological data can be obtained from the European Centre for Medium-Range Weather Forecasts (ECMWF) (http://apps.ecmwf.int/datasets/data/interim-full-daily/levtype=sfc/). The Multi-Source Weighted-Ensemble Precipitation (MSWEP) data can be downloaded from the website: http://www.gloh2o.org/.

5   The Ramankutty potential land-cover can be obtained from the website: https://nelson.wisc.edu/sage/data-and-models/global-potential-vegetation/index.php. The current cropland and pasture map can be obtained from EarthStat (http://www.earthstat.org/data-download/). Land Cover Type Climate Modeling Grid (CMG) MCD12C1 International Geosphere Biosphere Program (IGBP) land classification created from Terra and Aqua Moderate Resolution Imaging Spectroradiometer (MODIS) data for the year 2005 can be downloaded at https://modis.gsfc.nasa.gov/data/da Monthly irrigated rice and irrigation non-rice crops were obtained from the data set of Monthly Irrigated and Rainfed Crop Areas around the

10   year 2000 (MIRCA2000) V1.1. and can be downloaded at http://www.uni-frankfurt.de/45218031.

*Author contributions.* Research was conceived by L.W-E., R.J.v.d.E., P.W.K., H.H.G.S., and L.J.G.. I.F. contributed ideas for analyses. L.W-E. carried out the model simulations, analysed the data, and wrote the paper with input from all authors.

*Competing interests.* The authors declare that they have no conflict of interest.

*Acknowledgements.* We thank Victor Galaz, Chandrakant Singh, and two anonymous reviewers for providing feedback on the manuscript.

15   L.W-E., P.W.K, and L.J.G, are funded by The Swedish Research Council Formas (grant number 1364115). L.W-E. is also funded by the Japan Society for the Promotion of Science (JSPS). R.J.v.d.E. received funding from the European Union Seventh Framework Programme (FP7/2007–2013, grant agreement no. 603608). I.F. receives financial support from The Stordalen Foundation. I.F. and L.J.G. are also supported by The Swedish foundation for strategic environmental research (MISTRA).

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
