# Peer review of "Remote land use impacts on river flows through atmospheric teleconnections"

_Hydrology and Earth System Sciences, 2017_

## Referee Comment (RC1) · Anonymous Referee #1 · 20 Oct 2017

Anomalous review for "Remote land use impacts on river flows through atmospheric Teleconnections"

This study explores how the local and remote land use changes (LUC) might affect the river runoff. This study explores how the land-atmosphere interactions due to LUC can affect the rive runoff changes. Unlike previous study on the impacts of LUC on the river runoff, this study further considers the role of land-atmosphere interactions due to LUC in affecting the precipitation, then the river runoff. The main conclusion is that when we study the LCU effects on the river runoff, we cannot exclude the effect from Land-Atmosphere Interactions, and consideration of terrestrial moisture recycling is essential for understanding the changes in river runoff. Overall, the findings presented in this paper may be of interest to the community; however, there are several aspects that need to be addressed before the paper is accepted.

1.  Most of the regions show decreases in evaporation. It seems like the effect from irrigation is rather small? Any reasons? Also, I suggest to include a figure showing the irrigation water amount applied in this study that might be useful.

2.  P2L31: "no studies have quantified the magnitude of LUC impacts on P or Q".

    There are several studies have quantified the magnitude of LUC on P. The definition of LUC is rather broad, irrigation, deforestation, urbanization, so please carefully check those literatures.

3.  P3L25 "P differences between model iterations converges after about four simulations."

    Why after four simulations is choosing? Any particular physical meaning? Also, how long is the "four simulations"? 12 hours? (4 time steps?)

4.  P3L22-25 "P under potential land cover is obtained through a coupled model simulation. We use E output from STEAM in WAM-2layers, and iteratively adjust the current day P forcing to STEAM with the changes in P with terrestrial origin obtained by forward tracking continental moisture in WAM-2layers (SI Materials and Methods). P differences between model iterations converges after about four simulations."

    Based on this description of the "considering the L-A coupling", I guess this model does not take into account the effect the atmospheric nonlinearity nor the atmospheric circulations changes. While the authors have nicely summarized the different approaches in exploring the effects of LUC in Table S1. However, can the authors elaborate this further? In other words, how different the result might be if we use the couple global climate model to conduct the similar LUC experiment?

5.  P4L25:" (i.e., meeting the convergence requirement of mean annual precipitation change < 1% and monthly precipitation change < 5 mm/month in every grid cell"

    The threshold value should be clearly mentioned regarding the reason to choose such values of 1% and 5mm/month. Are there any sensitivity tests to achieve such values?

6. P5L13: "Our results show that human LUC (from potential land cover to current land use) (Fig. 1a) has"

   Be clearly on how to obtain the difference (or anomaly). Is from the potential land cover minus current land use? Or vice versa? Please also indication in the caption of Figure 1.

7. Figure 3 is interesting. It might be nice to include the changes in P and E on the bar chart plots.

8. Some of the figures from the supporting information can be moved to the main content for the readers to read, and there are not many figures in the main content either at this draft.

9. The ocean's E seems to be fixed because of using the reanalysis product. So, will the LUC over land affect the ocean's E? IF yes, the ocean's responses are completely ignored in this study. The authors may want to elaborate this issue further on the discussion as well. Also, to what extent the results from this study may be altered after considering the ocean's effects?

10. Can the model simulate the surface temperature changes due to LUC? We usually can see the changes in surface temperature accompanying with changes in evaporation. It will be nice to show the figure of surface temperature changes as well.

---

## Author Comment (AC1) · 25 Dec 2017

**Response to Anonymous Referee #1's comments on manuscript hess-2017-494 (Remote land use impacts on river flows through atmospheric teleconnections)**

We would like to thank the anonymous review #1 for the suggestions, comments, and questions. They strongly help to increase the quality of our manuscript by letting us improve the explanation of our methods and the presentation of our results. We are currently reworking the manuscript to reflect the suggestions made by the reviewer. Please see our detailed response below (reviewer quotes in blue and italic).

*1. Most of the regions show decreases in evaporation. It seems like the effect from irrigation is rather small? Any reasons? Also, I suggest to include a figure showing the irrigation water amount applied in this study that might be useful.*

Our estimates of evaporation change are on the conservative side for both decreases and increases, but is in range with current model and observation-based estimates as shown in Table 1 below. We will include a table with detailed comparison in the revised manuscript as shown below.

*Table 1. Overview of studies of land-use induced changes in evaporation. Pure irrigation studies are included in this comparison with irrigation water consumption reported as "E increase".*

| Reference | Model; Prec. forcing | E decreases | | E increases | | Overall E change | |
|---|---|---|---|---|---|---|---|
| | | km$^3$ y$^{-1}$ | % | km$^3$ y$^{-1}$ | % | km$^3$ y$^{-1}$ | % |
| (Döll and Siebert, 2002) | WaterGAP; CRU | - | - | +1100 | - | - | - |
| (Gordon et al., 2005) | Reference E; UDEL | -3000 | - 4.5 | +2600 | +3.9 | - 400 | - 0.6 |
| (Rost et al., 2008) | LPJmL; HadCM3/CRU | -2360 | - 3.8 | +1325 | +2.2 | -1036 | - 1.7 |
| (Sterling et al., 2012) | ORCHIDEE; NgoDuc05 | - | - | - | - | - 3160 | - 5.6 |
| (Sterling et al., 2012) | Literature-GIS; - | -2800 | -4.2 | +750 | +1.1 | - 3500 | - 5.3 |
| (Wada et al., 2014) | PCR-GLOBWB; CRU | - | - | +1179 | - | - | - |
| (Wada et al., 2014) | PCR-GLOBWB; ERA-I | - | - | +1120 | - | - | - |
| (Wada et al., 2014) | PCR-GLOBWB; MERRA | - | - | +994 | - | - | - |
| (Wang-Erlandsson et al., 2014) | STEAM; ERA-I | - | - | +1134 | +1.5 | - | - |
| (Jägermeyr et al., 2016) | LPJmL; GPCC | - | - | +1268 | - | - | - |
| This study | STEAM; MSWEP | -2047 | -3.0 | +796 | +1.2 | - 1251 | - 1.8 |

*2. P2L31: "no studies have quantified the magnitude of LUC impacts on P or Q". There are several studies have quantified the magnitude of LUC on P. The definition of LUC is rather broad, irrigation, deforestation, urbanization, so please carefully check those literatures.*

We thank the reviewer for pointing out that the sentence formulation can easily be misunderstood, especially when taken out of the context. We found the formulation to be redundant upon revisiting the paragraph, and will replace the original paragraph:

"The previous studies that illustrated the importance of remote LUC for basin P and Q, did not systematically assess global effects of LUC on Q, or explore the interplay between LUC within and outside the river basin. These effects are important to disentangle since they can have profound water governance implications (for e.g., riparian water rights and transboundary river basin treaties).While there has been discussions of governance implications of land-atmosphere interactions purely based on atmospheric moisture fluxes between nation states (Keys et al., 2017; Dirmeyer et al., 2009; Ellison et al., 2017), no studies have quantified the magnitude of LUC impacts on P or Q, despite its high relevance for international water law and governance. Thus, there is a missing interdisciplinary bridge between understanding the role of land-atmosphere feedback over large distances and its importance for water governance at the basin scale."

by

"The previous studies that illustrated the importance of remote LUC for basin P and Q, did not examine the effect of taking moisture recycling into account for estimating LUC effects on Q, nor analyse the interplay between LUC within and outside the river basin. These effects are, however, important to disentangle since they can have profound water governance implications (for e.g., riparian water rights and transboundary river basin treaties) (Keys et al., 2017; Dirmeyer et al., 2009; Ellison et al., 2017). Thus, there is a missing interdisciplinary bridge between understanding the role of land-atmosphere feedback over large distances and its importance for water governance at the basin scale."

*3. P3L25 "P differences between model iterations converges after about four simulations." Why after four simulations is choosing? Any particular physical meaning? Also, how long is the "four simulations"? 12 hours? (4 time steps?)*

The $P$ difference convergence requirement is described at page 4 line 25-26 and in Figure S9, and is less than 1 % per year and 5 mm/month in any grid cell. This simply means that an extra iteration would not yield very different results, and thus does not in any substantial way alter our findings and conclusions.

With our land-use change scenario, four simulation iterations were required to fulfil the convergence criteria at the global scale. The number of simulations can, thus, somewhat vary depending on e.g., the degree of land-use change, and the spatial or temporal coverage considered.

One coupled STEAM-WAM2layers simulation covers the entire time period 2000-2013 plus spin-up. Following the reviewer's suggestion, we will move Fig S9 to the main manuscript to prevent the misunderstanding concerning simulation length.

*4. P3L22-25 "P under potential land cover is obtained through a coupled model simulation. We use E output from STEAM in WAM-2layers, and iteratively adjust the current day P forcing to STEAM with the changes in P with terrestrial origin obtained by forward tracking continental moisture in WAM-2layers (SI Materials and Methods). P differences between model iterations converges after about four simulations." Based on this description of the "considering the L-A coupling", I guess this model does not take into account the effect the atmospheric nonlinearity nor the atmospheric circulations changes. While the authors have nicely summarized the different approaches in exploring the effects of LUC in Table S1. However, can the authors elaborate this further? In other words, how different the result might be if we use the couple global climate model to conduct the similar LUC experiment?*

The reviewer's question about the outcomes of climate models is interesting, but difficult mainly due to low signal-to-noise ratio and considerable inter-model differences in climate model simulations. As explained in "Introduction" and Table S1, not only are coupled global climate models conducting similar LUC experiment likely to give a low signal-to-noise ratio, but comprehensive climate model inter-comparison projects (Aloysius et al., 2016; Pitman et al., 2009) have also shown that precipitation changes from land-use change are highly inconsistent between models. Thus, for these reasons, it is difficult to speculate about how our results would potentially change by the use of climate models. To better convey this difficulty to the reader, we will move some of the discussions on climate models from the SI to the main manuscript

*5. P4L25:" (i.e., meeting the convergence requirement of mean annual precipitation change < 1% and monthly precipitation change < 5 mm/month in every grid cell" The threshold value should be clearly mentioned regarding the reason to choose such values of 1% and 5mm/month. Are there any sensitivity tests to achieve such values?*

We thank the reviewer for this question, but we are not entirely sure about what is meant. The convergence criterion is a sort of sensitivity test in itself. It means that the annual precipitation change is <1% and less than 5 mm/month when an additional iteration would be done.

*6. P5L13: "Our results show that human LUC (from potential land cover to current land use) (Fig. 1a) has" Be clearly on how to obtain the difference (or anomaly). Is from the potential land cover minus current land use? Or vice versa? Please also indication in the caption of Figure 1.*

We thank the reviewer for this comment which helps us better explain our analyses. However, because land-use in our case are represented by categories, rather than a scaled number (e.g., the case of tree cover percentage), the land-use change map is not obtained through subtraction. We simply show the grid cells where current land-use differ from potential land cover in Fig. 1a. Following the reviewer's comment, we intend to rephrase the caption in Figure 1 by changing the formulation from

"Differences between the current land-use and the potential land cover scenario. Changes in **a**, land use (only shifts in grid cell dominant land-use types are shown), **b**, …"

to

"Changes in land-use and water flows resulting from the replacement of the potential land cover scenario with the current land-use scenario. Changes in **a**, land use (current land use is shown, with grid cells without major land-use change masked out), **b**, …"

The sentence at P5L13 describes how the direction of land-use change (i.e., from potential to current) has altered the water cycle, which we think is clear in its context: "Our results show that human LUC (from potential land cover to current land use) (Fig. 1a) has led to reductions in *E* and *P*, and to increases in *Q*, in most regions (Fig. 1b-d)."

*7. Figure 3 is interesting. It might be nice to include the changes in P and E on the bar chart plots.*

We thank the reviewer for this comment, but we are not sure what is meant. The changes in *P* and *E* are already included in the bar chart plot in Fig 3a.

*8. Some of the figures from the supporting information can be moved to the main content for the readers to read, and there are not many figures in the main content either at this draft.*

Our feeling was that many of the figures in the SI constitute supplementary information do not contribute further to the key conclusions. However, we would be happy to move some of the figures to the main manuscript. Following the referee's previous suggestions, we plan to move Fig S9 to the main manuscript, to clarify the model coupling and convergence. We also plan to move Table S1 to

the main manuscript section to give a more comprehensive background to the rationale for using the current model coupling method.

*9. The ocean's E seems to be fixed because of using the reanalysis product. So, will the LUC over land affect the ocean's E? IF yes, the ocean's responses are completely ignored in this study. The authors may want to elaborate this issue further on the discussion as well. Also, to what extent the results from this study may be altered after considering the ocean's effects?*

The land-use change over land may affect above ocean processes mainly through modification of the energy balance and circulation in monsoon regions, which as stated, are not accounted for in the study. At P2L25, we refer to the PhD thesis of (Tuinenburg, 2013), which specifically examined the role of circulation change in estimates of land-use change effects on precipitation. In our current model set-up, an increase in irrigation leads to an increase in regional precipitation. Tuinenburg (2013) showed how precipitation might actually decrease by taking monsoon circulation response into account. Fully coupled ocean-atmosphere global climate models further increases the noise in simulation results (see also our response to Reviewer comment 4).

Changes in fresh water discharge to ocean might have implications ocean circulation and climate as studies of for example river discharge to the Arctic Ocean shows (Peterson, 2002, 2006). However, moisture recycling's buffering effect (which mitigates river flow changes), should in fact have a mitigating effect on the ocean's response to fresh water inflow. Otherwise, precipitation over ocean can influence ocean salinity (IPCC, 2013) and precipitation patterns over land can be influenced by sea surface temperature (Xie et al., 2010), but we consider this outside the scope of our study and likely to be of minor importance for the research questions that we address. More generally, land-use change over land is connected to the biogeochemical cycle. As part of the climate system, perturbation of the e.g., the carbon balance and land surface roughness through land-use change may also be connected to the ocean's surface temperature and wind speeds that might affect ocean evaporation feedback (Trenberth, 2011; van der Werf et al., 2009).

We suggest to add a discussion of these issues to the revised manuscript, although we would like to refrain from speculating about how "the extent the results from this study may be altered" (please also see our response to Reviewer comment 4).

*10. Can the model simulate the surface temperature changes due to LUC? We usually can see the changes in surface temperature accompanying with changes in evaporation. It will be nice to show the figure of surface temperature changes as well.*

Changes in surface temperature are not simulated. Changes in potential evaporation (Penman-Monteith equation) are taken into account through changes in land parameters.

**Bibliography**

Aloysius, N. R., Sheffield, J., Saiers, J. E., Li, H. and Wood, E. F.: Evaluation of historical and future simulations of precipitation and temperature in central Africa from CMIP5 climate models, J. Geophys. Res. Atmos., 121(1), 130–152, doi:10.1002/2015JD023656, 2016.

Döll, P. and Siebert, S.: Global modeling of irrigation water requirements, Water Resour. Res., 38(4),

8-1-8–10, doi:10.1029/2001WR000355, 2002.

Gordon, L. J., Steffen, W., Jönsson, B. F., Folke, C., Falkenmark, M., Johannessen, A. and Johannessen, Å.: Human modification of global water vapor flows from the land surface., Proc. Natl. Acad. Sci. U. S. A., 102(21), 7612–7617, doi:10.1073/pnas.0500208102, 2005.

IPCC: Climate Change 2013: The Physical Science Basis,Contribution of Working Group I to the Fifth Assessment Report of the Intergovernmental Panel on Climate Change., 2013.

Jägermeyr, J., Gerten, D., Schaphoff, S., Heinke, J., Lucht, W. and Rockström, J.: Integrated crop water management might sustainably halve the global food gap, Environ. Res. Lett., 11(2), 25002, doi:10.1088/1748-9326/11/2/025002, 2016.

Peterson, B. J.: Increasing River Discharge to the Arctic Ocean, Science, 298(5601), 2171–2173, doi:10.1126/science.1077445, 2002.

Peterson, B. J.: Trajectory Shifts in the Arctic and Subarctic Freshwater Cycle, Science, 313(5790), 1061–1066, doi:10.1126/science.1122593, 2006.

Pitman, A. J., de Noblet-Ducoudré, N., Cruz, F. T., Davin, E. L., Bonan, G. B., Brovkin, V., Claussen, M., Delire, C., Ganzeveld, L., Gayler, V., van den Hurk, B. J. J. M., Lawrence, P. J., van der Molen, M. K., Müller, C., Reick, C. H., Seneviratne, S. I., Strengers, B. J. and Voldoire, A.: Uncertainties in climate responses to past land cover change: First results from the LUCID intercomparison study, Geophys. Res. Lett., 36(14), 1–6, doi:10.1029/2009GL039076, 2009.

Rost, S., Gerten, D. and Heyder, U.: Human alterations of the terrestrial water cycle through land management, Adv. Geosci., 18(18), 43–50, doi:10.5194/adgeo-18-43-2008, 2008.

Sterling, S. M., Ducharne, A. and Polcher, J.: The impact of global land-cover change on the terrestrial water cycle, Nat. Clim. Chang., 3(4), 385–390, doi:10.1038/nclimate1690, 2012.

Trenberth, K. E.: Changes in precipitation with climate change, Clim. Res., 47(1), 123–138, doi:10.3354/cr00953, 2011.

Tuinenburg, O. A.: Atmospheric Effects of Irrigation in Moonsoon Climate: The Indian Subcontinent, Wageningen University., 2013.

Wada, Y., Wisser, D. and Bierkens, M. F. P.: Global modeling of withdrawal, allocation and consumptive use of surface water and groundwater resources, Earth Syst. Dyn., 5(1), 15–40, doi:10.5194/esd-5-15-2014, 2014.

Wang-Erlandsson, L., van der Ent, R. J., Gordon, L. J. and Savenije, H. H. G.: Contrasting roles of interception and transpiration in the hydrological cycle – Part 1: Temporal characteristics over land, Earth Syst. Dyn., 5(2), 441–469, doi:10.5194/esd-5-441-2014, 2014.

van der Werf, G. R., Morton, D. C., DeFries, R. S., Olivier, J. G. J., Kasibhatla, P. S., Jackson, R. B., Collatz, G. J. and Randerson, J. T.: CO2 emissions from forest loss, Nat. Geosci., 2(11), 737–738, doi:10.1038/ngeo671, 2009.

Xie, S. P., Deser, C., Vecchi, G. a, Ma, J., Teng, H. Y. and Wittenberg,  a T.: Global Warming Pattern Formation: Sea Surface Temperature and Rainfall, J. Clim., 23(4), 966–986, doi:10.1175/2009jcli3329.1, 2010.

---

## Referee Comment (RC2) · Anonymous Referee #2 · 13 Mar 2018

**Review: Remote land use impacts on river flows through atmospheric Teleconnections**

The paper studies the interaction between land use changes inside and outside of a basin on the river flow by coupling of a hydrological model (STEAM) with an atmospheric moisture tracking model (WAM-2layers). The study comes to the conclusion that depending on the region extra-basin land use changes can strongly affect river flow inside a basin.

The topic of land use change impact through atmospheric teleconnection is very interesting and important to the hydrological science community as well as for water governance. The method of coupling an atmospheric model with a hydrological model in both directions (not as usual in one direction) is very promising.

Main suggestions:

- Some of the descriptions, figures in the supplement should be moved to the main paper.
- The focus of basins should be moved from basins with high absolute delta Q (Amazon, Congo, Ob etc.) to basins with high % change (Indus, Zambezi, Odra)
- The role of TMR should be addressed regionally. To refer to the title the differences between having the teleconnection in or not should be pointed out. What are the basins affected by TMR most and why?

Some other aspects have to be addressed:

P2 L31: ".. no studies .." about P I do not know, but LUC on Q a lot: See the https://www.isimip.org/ project deals also with impact of human interactions on water balance variables.

P3 L14: on long term: q = P – ETact; where ETact is actual evapotranspiration with the parts mentioned before. Further on we talk about actual ET?

P4 L15 and S8: GRDC runoff data are used for verification. GRDC data are not referred and not explained see: http://www.bafg.de/GRDC/EN/03_dtprdcts/33_CmpR/unh_grdc_node.html). Also the data source is from 2002 (and not really "observed" runoff). Maybe a comparison with recent modelling results is more appropriated (see: https://www.isimip.org/outputdata/), also because the MSWEP precipitation data is more inline with the WATCH WDFEI dataset than the one used in 2002. But here the important part is to show that the hydrological model is more or less ok. Therefore a rough comparison might be ok. Maybe adding the explanation of the supplement is enough.

P4 L20: E,pot might be misleading because it is normally used for potential Evaporation

P4 L22: Why Fig S8 as reference?

P6 Figure 1a: Irrigated crop (orange) is hard to distinguish from rainfed crops (red). From this map there is hardly any irrigation in Spain or Italy (even if it is the main land cover change). What are the green dots?. A land use change map which indicated from with land cover to which would be helpful e.g.

Forest – pasture. Because it makes a difference if you change from Forest – pasture or from shrubland - pasture

P6 Figure 1b-d: Quite a drastic change. E.g. for Zambezi that is more than 100% (more than indicated in S1). Fig 1a-c are cell values.

Figure 1d are summed up for the catchment at the outlet and then again put on the whole catchment. Maybe choose another way to show this, e.g. as points. Or show this as percentage as in S1h. Also because a change of 1000 m3/s in the Amazonas (or Congo) is close to nothing while for the Elbe River it is a lot.

P6 L1-8: This part is also interesting. But it would be good to have a global map here instead only a description of some basins to see the region differences. Maybe a map of absolute delta Q and delta Q/Q. Fig S1h shows the interesting basins like Odra, Indus, Colorado, Niger, Zambezi

P7 Figure 2: This is a necessary figure to show that the high values in fig 1 are well based inside other literature. Please put a table or part of it or another way of displaying results here in order to make it independent from the supplementary.

P8 Much more interesting than the Congo (less than 5% change) is the Zambezi with almost 100% change.

P8 Fig3: The y-axis is not only hydrological flow [m3/s]

P10 L 21ff:

- Precipitation and evaporation over the sea is not in
- Changes in atmospheric circulation is not in. it seems the model assumes the same patterns (see S1e)?
- Are you sure that dams are not indirectly in. e.g. Zambezi big dams as major land cover change from shrubland to open water?

---

## Author Comment (AC2) · 27 Mar 2018

**Response to Anonymous Referee #2's comments on manuscript hess-2017-494 (Remote land use impacts on river flows through atmospheric teleconnections)**

We would like to thank anonymous review #2 for the suggestions, comments, and questions. They strongly help to increase the quality of our manuscript by letting us add additional river basins of relevance to the analyses, sharpen the presentation of our results and delve deeper in the discussions. Please see our detailed response below (reviewer quotes in blue and italic).

*Main suggestions:*

*- Some of the descriptions, figures in the supplement should be moved to the main paper.*

Thank you for this suggestion. Following also the comments from referee #1, we plan to move Fig S9 to the main manuscript, to clarify the model coupling and convergence. We also plan to move Table S1 to the main manuscript section to give a more comprehensive background to the rationale for using the current model coupling method. To better convey this difficulty to the reader, we will also move some of the discussions on climate models from the supplement to the main manuscript.

*- The focus of basins should be moved from basins with high absolute delta Q (Amazon, Congo, Ob etc.) to basins with high % change (Indus, Zambezi, Odra)*

We thank the referee for this valuable suggestion. Though we initially did not focus on the basins with high % change in imported precipitation, we acknowledge that big relative changes in river flows are likely to matter more from the perspective of the water users. Thus, we will add two large river basins with large relative river flow reductions (Yellow, Huai, Colorado) and three with large relative increases (Zambezi, Niger) for which river flow (i.e., P-E) simulated by STEAM matches well with river flow data. This addition will modify Fig 3 as well as the country level analyses and related discussions.

*- The role of TMR should be addressed regionally. To refer to the title the differences between having the teleconnection in or not should be pointed out. What are the basins affected by TMR most and why?*

Thank you for your comment. However, the analyses in this paper estimated the human land-use change effect in combination with TMR, and Fig 3 lists the basins most strongly affected by TMR given the land-use change scenario employed. Thus, the effect of TMR observed is highly dependent on prevailing winds, hydroclimate, and orography (van der Ent et al., 2010, 2014) and the land-use change scenario analysed, as we also observe in the manuscript. We will make sure to elaborate a bit more about this in a revised version.

*Some other aspects have to be addressed:*

*P2 L31: ".. no studies .." about P I do not know, but LUC on Q a lot: See the https://www.isimip.org/ project deals also with impact of human interactions on water balance variables.*

We thank both reviewer #1 and #2 for pointing out that the sentence formulation can easily be misunderstood, especially when taken out of the context. We found the formulation to be redundant upon revisiting the paragraph, and will replace the original paragraph P2L27-P2L34 by

"The previous studies that illustrated the importance of remote LUC for basin P and Q, did not examine the effect of taking moisture recycling into account for estimating LUC effects on *Q*, nor analyse the interplay between LUC within and outside the river basin. These effects are, however, important to disentangle since they can have profound water governance implications (for e.g., riparian water rights and transboundary river basin treaties) (Dirmeyer et al., 2009; Ellison et al., 2017; Keys et al., 2017). Thus, there is a missing interdisciplinary bridge between understanding the role of land-atmosphere feedback over large distances and its importance for water governance at the basin scale."

*P3 L14: on long term: q = P – ETact; where ETact is actual evapotranspiration with the parts mentioned before. Further on we talk about actual ET?*

I am not sure I understand the question correctly. To be clear in the manuscript, we will add the equation Q = P – E directly after the sentence ending at P3L15.

*P4 L15 and S8: GRDC runoff data are used for verification. GRDC data are not referred and not explained see: http://www.bafg.de/GRDC/EN/03_dtprdcts/33_CmpR/unh_grdc_node.html). Also the data source is from 2002 (and not really "observed" runoff). Maybe a comparison with recent modelling results is more appropriated (see: https://www.isimip.org/outputdata/), also because the MSWEP precipitation data is more inline with the WATCH WDFEI dataset than the one used in 2002. But here the important part is to show that the hydrological model is more or less ok. Therefore a rough comparison might be ok. Maybe adding the explanation of the supplement is enough.*

We thank the reviewer for this insightful comment. GRDC was only referred to in the Supplementary Information, and a reference was indeed omitted in the main text body. In the revision, we will make sure that GRDC data is referred to and explained in the Data section in the main manuscript where it is first mentioned. It is as the referee points out, important to note that GRDC does not represent the best discharge data available. For example in the River Niger, the GRDC discharge is around 0.2 m/y (Fig S8), while station data at Lokoja station for 1980-2013 shows that it is about 0.07 m/y (Oyerinde et al., 2017).

*P4 L20: E,pot might be misleading because it is normally used for potential Evaporation*

Thank you for pointing this out. We will change $E_{pot}$ to $E_{pv}$ instead.

*P4 L22: Why Fig S8 as reference?*

Thank you for the sharp observation. We meant to refer to Fig S9.

*P6 Figure 1a: Irrigated crop (orange) is hard to distinguish from rainfed crops (red). From this map there is hardly any irrigation in Spain or Italy (even if it is the main land cover change). What are the green dots?. A land use change map which indicated from with land cover to which would be helpful e.g.Forest – pasture. Because it makes a difference if you change from Forest – pasture or from shrubland – pasture*

Great comments. We will change the colour scheme in Fig 1a, so the differences between irrigated and rainfed will become easier to distinguish as well as add categories of in terms of the original land cover (e.g., "forest to pasture"). The map shows the main difference between current and potential land cover, and although irrigation is high in Spain and Italy, the rainfed agriculture still dominates. We will consider showing the percentage of rainfed and irrigated cropland separately in the revised manuscript. The green dots are increase in forest cover and this was not included in the legend, due to the difficulties of seeing the fine changes (i.e., tiny dots) in a world map. We will add a bar figure summarizing the land-use changes in terms original to current (e.g., total area of forest changed to pasture etc.).

*P6 Figure 1b-d: Quite a drastic change. E.g. for Zambezi that is more than 100% (more than indicated in S1). Fig 1a-c are cell values.*

*Figure 1d are summed up for the catchment at the outlet and then again put on the whole catchment. Maybe choose another way to show this, e.g. as points. Or show this as percentage as in S1h. Also because a change of 1000 m3/s in the Amazonas (or Congo) is close to nothing while for the Elbe River it is a lot.*

We will modify the subheading of Figure 1d so that it clearly says "River flow change at outlet" as well as move Fig S1h from the supplements to the main manuscript.

*P6 L1-8: This part is also interesting. But it would be good to have a global map here instead only a description of some basins to see the region differences. Maybe a map of absolute delta Q and delta Q/Q. Fig S1h shows the interesting basins like Odra, Indus, Colorado, Niger, Zambezi*

Absolute difference in river flows is shown in map format in Fig 1d, and we will move the map of relative differences of river flows from the supplements Fig S1h to the main manuscript. Fig 3a summarises both absolute and relative river flows by basin. In the revised manuscript, we will add the basins with large relative river flow change to the figures.

*P7 Figure 2: This is a necessary figure to show that the high values in fig 1 are well based inside other literature. Please put a table or part of it or another way of displaying results here in order to make it independent from the supplementary.*

Good point, we will add a legend to Fig 2 that lists all cited literature.

*P8 Much more interesting than the Congo (less than 5% change) is the Zambezi with almost 100% change.*

Excellent point. We will add Zambezi (among others) to the analyses.

*P8 Fig3: The y-axis is not only hydrological flow [m3/s]*

The figure has two y-axis, m3/s to the left and % to the right. We will add a remark about this in the caption.

*P10 L 21ff:*

*- Precipitation and evaporation over the sea is not in*

Yes, it is true that the ocean precipitation and evaporation is not accounted for. Following also the comments from referee #1, we suggest to add a discussion of these issues to the revised manuscript.

The land-use change over land may affect above ocean processes mainly through modification of the energy balance and circulation in monsoon regions, which as stated, are not accounted for in the study. At P2L25, we refer to the PhD thesis of (Tuinenburg, 2013), which specifically examined the role of circulation change in estimates of land-use change effects on precipitation. In our current model set-up, an increase in irrigation leads to an increase in regional precipitation. Tuinenburg (2013) showed how precipitation might actually decrease by taking monsoon circulation response into account. Fully coupled ocean-atmosphere global climate models further increases the noise in simulation results.

Changes in fresh water discharge to ocean might have implications ocean circulation and climate as studies of for example river discharge to the Arctic Ocean shows (Peterson, 2002, 2006). However, moisture recycling's buffering effect (which mitigates river flow changes), should in fact have a mitigating effect on the ocean's response to fresh water inflow. Otherwise, precipitation over ocean can influence ocean salinity (IPCC, 2013) and precipitation patterns over land can be influenced by sea surface temperature (Xie et al., 2010), but we consider this outside the scope of our study and likely to be of minor importance for the research questions that we address. More generally, landuse change over land is connected to the biogeochemical cycle. As part of the climate system, perturbation of the e.g., the carbon balance and land surface roughness through land-use change may also be connected to the ocean's surface temperature and wind speeds that might affect ocean evaporation feedback (Trenberth, 2011; van der Werf et al., 2009) .

*- Changes in atmospheric circulation is not in. it seems the model assumes the same patterns (see S1e)?*

Yes, this is explicitly stated in this section. By moving some of the discussions on climate models from the supplement to the main manuscript, we hope this will become clearer as well. See also our reply to Reviewer #1's comment no. 4.

*- Are you sure that dams are not indirectly in. e.g. Zambezi big dams as major land cover change from shrubland to open water?*

Good point. We will include a description of this type of indirect land-use change in connection to the bar plot of land-use change we plan to add to the Result discussion, as well as in the Discussion section.

**References**

Dirmeyer, P. A., Brubaker, K. L. and DelSole, T.: Import and export of atmospheric water vapor between nations, J. Hydrol., 365(1–2), 11–22, doi:10.1016/j.jhydrol.2008.11.016, 2009.

Ellison, D., Morris, C. E., Locatelli, B., Sheil, D., Cohen, J., Murdiyarso, D., Gutierrez, V., Noordwijk, M. van, Creed, I. F., Pokorny, J., Gaveau, D., Spracklen, D. V., Tobella, A. B., Ilstedt, U., Teuling, A. J., Gebrehiwot, S. G., Sands, D. C., Muys, B., Verbist, B., Springgay, E., Sugandi, Y. and Sullivan, C. A.: Trees, forests and water: Cool insights for a hot world, Glob. Environ. Chang., 43, 51–61, doi:10.1016/j.gloenvcha.2017.01.002, 2017.

van der Ent, R. J., Savenije, H. H. G. G., Schaefli, B. and Steele-Dunne, S. C.: Origin and fate of atmospheric moisture over continents, Water Resour. Res., 46(9), 1–12, doi:10.1029/2010WR009127, 2010.

van der Ent, R. J., Wang-Erlandsson, L., Keys, P. W. and Savenije, H. H. G.: Contrasting roles of interception and transpiration in the hydrological cycle – Part 2: Moisture recycling, Earth Syst. Dyn., 5(2), 471–489, doi:10.5194/esd-5-471-2014, 2014.

IPCC: Climate Change 2013: The Physical Science Basis,Contribution of Working Group I to the Fifth Assessment Report of the Intergovernmental Panel on Climate Change., 2013.

Keys, P. W., Wang-Erlandsson, L., Gordon, L. J., Galaz, V. and Ebbesson, J.: Approaching moisture recycling governance, Glob. Environ. Chang., 45, 15–23, doi:10.1016/j.gloenvcha.2017.04.007, 2017.

Oyerinde, G. T., Fademi, I. O. and Denton, O. A.: Modeling runoff with satellite-based rainfall estimates in the Niger basin, edited by M. Tejada Moral, Cogent Food Agric., 3(1), doi:10.1080/23311932.2017.1363340, 2017.

Peterson, B. J.: Increasing River Discharge to the Arctic Ocean, Science, 298(5601), 2171–2173, doi:10.1126/science.1077445, 2002.

Peterson, B. J.: Trajectory Shifts in the Arctic and Subarctic Freshwater Cycle, Science, 313(5790), 1061–1066, doi:10.1126/science.1122593, 2006.

Trenberth, K. E.: Changes in precipitation with climate change, Clim. Res., 47(1), 123–138, doi:10.3354/cr00953, 2011.

Tuinenburg, O. A.: Atmospheric Effects of Irrigation in Moonsoon Climate: The Indian Subcontinent, Wageningen University., 2013.

van der Werf, G. R., Morton, D. C., DeFries, R. S., Olivier, J. G. J., Kasibhatla, P. S., Jackson, R. B., Collatz, G. J. and Randerson, J. T.: CO2 emissions from forest loss, Nat. Geosci., 2(11), 737–738, doi:10.1038/ngeo671, 2009.

Xie, S. P., Deser, C., Vecchi, G. a, Ma, J., Teng, H. Y. and Wittenberg,  a T.: Global Warming Pattern Formation: Sea Surface Temperature and Rainfall, J. Clim., 23(4), 966–986, doi:10.1175/2009jcli3329.1, 2010.

---

## Author Response (AR2)

24 July 2018

Dear Editor,

We would like to thank you for the opportunity to resubmit a revised version of the manuscript entitled "Remote land use impacts on river flows through atmospheric teleconnections". We are grateful for the constructive comments and feedback from the two reviewers, which we believe have helped improve the manuscript in terms of clarity, structure, and substance. The additional analyses of five river basins with relatively low river flows, as suggested by one of the reviewers, particularly help shed light on the role and relevance of accounting for moisture recycling in water scarce basins.

In response to the comments of reviewer #1 and #2, we have made the following revisions to the manuscript:

- Throughout the manuscript
  - Replaced subscript $_{pot}$ with $_{pv}$ to stand for potential vegetation scenario. (In response to Reviewer #2 comment on P4 L20.).
  - Corrected cross-references where necessary (In response to Reviewer #2 comment on P4 L20.).
- Introduction
  - P.2: clarified how this study builds upon previous studies. (In response to Reviewer #1 comment 2, and Reviewer #2 comment on P2 L31.)
- Methods
  - Moved most of the methods descriptions from Supplementary Information to the main manuscript, including moving figure illustrating the model coupling procedure (now Fig. 1) to the main manuscript. (In response to Reviewer #1 comment 3.)
  - Added figure of land-use change scenarios and improved colour scheme (the new Fig. 2), as well as caption. (In response to Reviewer #1 comment 6, and Reviwer #2 comment on P6 Figure 1a.)
  - Improved the explanation of the water balance equation Q = P - E  (In response to Reviewer #2 comment on P3 L14.).
- Results
  - Moved maps of actual hydrological flows and precipitation recycling ratio to the main manuscript (new Fig. 3) from the SI.
  - Moved relative changes in hydrological flows to the main manuscript (new Fig.4) from the SI. Added absolute change in precipitation recycling ratio (Fig. 4e) for completeness. Clarified that river flow changes refer to "river flows changes at outlet". (In response to Reviewer #2 comment on P6 Figure 1b-d, and P6 L1-8)
  - Added literature sources of the now Fig. 5 directly in the figure, so readers no longer need to consult the SI for this information. (In response to Reviewer #2 comment on P7 Figure 2.)
  - Added new figure (Fig. 6) that shows the effect of TMR on river flows globally, and description of this figure in the text. (This addition was not explicitly requested by the reviewers, nor included in the earlier authors' comments. Nevertheless, during the implementation of the revisions, we found it to partly address Reviewer #2 main suggestion 3.)

- o Added five basins (Zambezi, Niger, Yellow, Huai, Colorado) to Fig. 7 and added a sentence in the figure caption to notify readers that there are two axes in the figure. (In response to Reviewer #2 main suggestion 2, comment on P8, and P8 Fig3.)
  - o Moved figures of precipitation- and evaporationsheds and their changes (new Fig. 8 and 9) of four basins (Amazon, Yangtze, Yenisei, Niger) from the SI to the main paper, and added descriptions in the text accordingly.
  - o Moved figure of nation influence in river flows (new Fig 10) of all 15 basins from the SI to the main paper, and added descriptions in the text accordingly.
- • Discussion
  - o Added discussion of the study's limitations in terms of circulation, ocean feedback, and dams to the Limitations section. (In response to Reviewer #1 comment 9, Reviewer #2 comment on P10 L 21ff, and unnumbered comment on atmospheric circulation and dams.)
  - o Moved substantial information regarding alternative methods to investigate land-rainfall-river flow feedback from the SI to the Limitations subsection. (In response to Reviewer #1 comment 4.)
  - o Added recent references.
- • Supplementary Information
  - o Information moved from SI to the main manuscript as specified above. (Also in response to Reviewer #1 comment 8, and Reviewer #2 main suggestion 1.)
  - o Explained the limitation of the GRDC dataset in the caption of Fig. S2. (In response to Reviewer #2 comment on P4 L15 and S8)
  - o Ordered the SI figures and tables in the order they are mentioned in the main manuscript. (Previously, there were two sections in SI: "Figures and Tables" and "Methods".)
  - o Added Fig. S1 that explains the change in parameterisation approach in STEAM for this study. (Not suggested by reviewers, but we found it relevant during the revision process.)
  - o Added Table S3 that provides an overview of other studies' estimates of land-use change induced increases and decreases in evaporation in absolute and relative terms. (In response to Reviewer #1 comment 1.)
  - o Added Fig. S3 that illustrates the land-cover and land-use scenarios used in the study. (In response to Reviewer #1 comment 6.)
  - o Added information of four of the new additional five basins (Zambezi, Yellow, Huai, Colorado) to Fig. S4-S7. (In response to Reviewer #2 main suggestion 2, and comment on P8.)

In a few instances, the reviewers made suggestions we did not agree with or assume are misunderstandings, as we also explained in the authors' comments posted in Interactive Discussion. In summary, these include:

- • Reviewer #1 comment 5 posed the question whether "there are sensitivity tests to achieve" the convergence requirements values used for determining the number of model coupling iterations. We clarified in our earlier author's comment that the convergence criterion can be seen as "a sort of sensitivity test in itself". We hope that the current revisions in the Methods section and improved explanation of our model coupling procedure will make this point clearer.

- Reviewer #1 comment 7 suggests including changes in P and E on the bar chart plots, which are already included (Fig. 7), and therefore did not require any revisions.
- Reviewer #1 comment 10 suggest showing surface temperature changes, which is not simulated in this study.

Full details of the revisions are shown in the marked-up manuscript.

We hope that the revised manuscript is now suitable for publication in HESS. Thank you for your consideration.

Sincerely yours,

Lan Wang-Erlandsson on behalf of the authors

[revised manuscript text omitted]

The pattern of overlapping precipitation- and evaporationsheds illustrated in Fig. 7b and moderated by wind directions can also be clearly seen in the basin specific precipitation- and evaporationshed maps (Fig. 8). In the Amazon (Fig. 8a,b), the moisture arrives from the east, is stopped up by the Andes, and changes direction towards the southeast. The hotspot of precipitation source and sink within the Amazon basin do not overlap, with major moisture providing spots located along the northeastern border and the major moisture receiving spots located along the Andes in the west. In the Yangtze (Fig. 8c,d), the moisture comes from a large area in the south, and leaves in the direction of Japan in a relatively narrow band. In the Yenisei (Fig. 8e,f), the moisture follows the westerlies, coming in straight from the west, and leaving straight towards the east. In the Niger (Fig. 8g,h), the moisture is mostly supplied from the east from terrestrial areas, and flows towards the west into the Atlantic. For precipitation- and evaporationsheds of other basins, see Fig. S4 and S6 respectively.

While changes in precipitation- and evaporationsheds are conditioned by the original moisture flows, the resulting pattern ultimately depend on the distribution of LUC-induced hydrological change (compare Fig. 8 and 9). For example, although the Amazon precipitationshed is weak over Africa (Fig. 8a), the precipitationshed change is in fact relatively strong there due to strong LUC-induced hydrological change (Fig. 9a). In other cases, aggregated changes in Fig. 7 hide spatially heterogeneous increases and decreases in moisture flows. For example, agricultural activities and irrigation in India, the Sahel, and regions around the Nile increase moisture supply to the Yangtze, Yensisei, and Niger basins and offset deforestation induced moisture supply decrease elsewhere (Fig. 9c,e,g). For changes in precipitation- and evaporationsheds of other basins, see Fig. S5 and S7 respectively.

**3.4 Attributing influence on $\Delta Q$ to nations**

Typically, TMR attributes LUC influence on $Q$ (methods described in Sect. 2.3.2) to a larger number of nations than when only basin boundaries are considered (Fig.  10). In several of the studied basins, (such as the Amazon, Congo, Volga, Ob, Yenisei, and Niger basins, see Fig. 10a,b,e,f,i,l), the share of nations contributing less than 5 % to $\Delta Q$  grows considerably when TMR is considered. In some cases, nations not considered a key influencer of $\Delta Q$, in fact influence $\Delta Q$  more than 5 % when TMR is accounted for: in Mekong, India is only an important influencer (10 % influence) when TMR is considered (Fig. 10g); in Yenisei, Mongolia falls below 5 % influence, while Kazakhstan (11 %) and China (6 %) climb considerably in influence (Fig. 10i); and in Niger basin, Sudan/South Sudan (8 %) and Niger (5 %) replace Ivory Coast and Guinea as important influencers (Fig. 10l). Notably, basins geographically confined within one nation can be influenced by LUC taking place in foreign nations. This is for example the case in Yangtze,  Yellow, and Huai, where irrigation in India increases the  basins' $P$ (Fig.  10d,m,n). The TMR lead to limited difference in nation influence only in the North American basins (Fig. 10h,o) and La Plata (Fig. 10c).

[revised manuscript text omitted]